# Clinical grade ACE2 as a universal agent to block SARS-CoV-2 variants

Vanessa Monteil[1] , Brett Eaton[2], Elena Postnikova[2], Michael Murphy[2], Benedict Braunsfeld[3] ,
Ian Crozier[4], Franz Kricek[5], Janine Niederhöfer[6], Alice Schwarzböck[6], Helene Breid[6],
Stephanie Devignot[1] , Jonas Klingström[7] , Charlotte Thålin[8], Max J Kellner[9,10], Wanda Christ[7],
Sebastian Havervall[8], Stefan Mereiter[9] , Sylvia Knapp[11] , Anna Sanchez Jimenez[6],
Agnes Bugajska-Schretter[6] , Alexander Dohnal[6] , Christine Ruf[5], Romana Gugenberger[6],
Astrid Hagelkruys[9] , Nuria Montserrat[12,13] , Ivona Kozieradzki[14] , Omar Hasan Ali[14] ,
Johannes Stadlmann[15] , Michael R Holbrook[2] , Connie Schmaljohn[2], Chris Oostenbrink[3] ,
Robert H Shoemaker[16] , Ali Mirazimi[1] , Gerald Wirnsberger[6,*] & Josef M Penninger[9,14,**]

## Abstract

The recent emergence of multiple SARS-CoV-2 variants has caused considerable concern due to both reduced vaccine efficacy and escape from neutralizing antibody therapeutics. It is, therefore, paramount to develop therapeutic strategies that inhibit all known and future SARS-CoV-2 variants. Here, we report that all SARS-CoV-2 variants analyzed, including variants of concern (VOC) Alpha, Beta, Gamma, Delta, and Omicron, exhibit enhanced binding affinity to clinical grade and phase 2 tested recombinant human soluble ACE2 (APN01). Importantly, soluble ACE2 neutralized infection of VeroE6 cells and human lung epithelial cells by all current VOC strains with markedly enhanced potency when compared to reference SARS-CoV-2 isolates. Effective inhibition of infections with SARS-CoV-2 variants was validated and confirmed in two independent laboratories. These data show that SARS-CoV-2 variants that have emerged around the world, including current VOC and several variants of interest, can be inhibited by soluble ACE2, providing proof of principle of a pan-SARS-CoV-2 therapeutic.

**Keywords** COVID-19; treatment; clinical trial; vaccine
**Subject Categories** Microbiology, Virology & Host Pathogen Interaction
**EMBO Mol Med (2022) e15230**

## Introduction

The emergence of SARS-CoV-2 has resulted in an unprecedented COVID-19 pandemic with dire economic, social, and health consequences for hundreds of millions of people. The initial step of SARS-CoV-2 infection is binding of the viral Spike protein to Angiotensin-converting enzyme 2 (ACE2) (Shang *et al*, 2020; Wang *et al*, 2020; Zhou *et al*, 2020), followed by proteolytic processing of the trimeric Spike (Benton *et al*, 2020; Hoffmann *et al*, 2020) and subsequent infection of target cells (Hu *et al*, 2021). Inhibition of Spike/ACE2 interaction is the fundamental principle for the activity of neutralizing antibodies induced by all current vaccines (Kyriakidis *et al*, 2021). Similarly, approved monoclonal antibodies act by blocking the

1 Unit of Clinical Microbiology, Karolinska Institutet and Karolinska University Hospital, Stockholm, Sweden
2 NIAID Integrated Research Facility at Fort Detrick (IRF-Frederick), Frederick, Maryland, USA
3 Institute for Molecular Modeling and Simulation, University of Natural Resources and Life Sciences (BOKU), Vienna, Austria
4 Clinical Research Monitoring Program Directorate, Frederick National Laboratory for Cancer Research, Frederick, Maryland, USA
5 NBS-C BioScience & Consulting GmbH, Vienna, Austria
6 invIOs, Vienna, Austria
7 Center for Infectious Medicine, Department of Medicine Huddinge, Karolinska Institutet, Stockholm, Sweden
8 Department of Clinical Sciences, Karolinska Institute Danderyd Hospital, Stockholm, Sweden
9 Institute of Molecular Biotechnology of the Austrian Academy of Sciences, Vienna, Austria
10 Vienna BioCenter PhD Program, Doctoral School of the University at Vienna and Medical, University of Vienna, Vienna, Austria
11 Department of Medicine 1, Laboratory of Infection Biology, Medical University of Vienna, Vienna, Austria
12 Pluripotency for Organ Regeneration, Institute for Bioengineering of Catalonia (IBEC), The Barcelona Institute of Science and Technology (BIST), Barcelona, Spain
13 Catalan Institution for Research and Advanced Studies (ICREA), Barcelona, Spain
14 Department of Medical Genetics, Life Sciences Institute, University of British Columbia, Vancouver, Canada
15 Institute of Biochemistry, Department of Chemistry, University of Natural resources and Life, Sciences (BOKU), Vienna, Austria
16 Chemopreventive Agent Development Research Group, Division of Cancer Prevention, National Cancer Institute, National Institutes of Health, Bethesda, Maryland, USA
*Corresponding author. Tel: +43 1 8656577300; E-mail: gwi@invios.com
**Corresponding author. Tel: +1 604 8270347; E-mail: josef.penninger@ubc.ca

interaction of the cell entry receptor ACE2 and the viral Spike protein (see https://www.covid19treatmentguidelines.nih.gov/ for further information). Thus, blocking Spike/ACE2 binding has become a central strategy of both vaccine design and multiple therapeutic approaches including ACE2-based therapeutics (Linsky *et al*, 2020; preprint: Svilenov *et al*, 2020; preprint: Hassler *et al*, 2021; Higuchi *et al*, 2021; Tanaka *et al*, 2021). This has created an intense research focus on the molecular details of these processes, thereby making the Spike/ACE2 interaction one of the best validated drug targets in medicine.

Both vaccines and antibody therapeutics have had an enormous impact and are a remarkable testament to the rapid translatability of basic research. However, although coronaviruses mutate less frequently, as compared to viruses like influenza, many variants of SARS-CoV-2 have emerged throughout the pandemic (Banerjee *et al*, 2021; Harvey *et al*, 2021) (see also WHO and CDC resources online). Some of these variants have been designated as variants of concern (VOCs) by the WHO because of their increased infectivity and transmissibility. Mutations in the viral Spike protein seem to be of particular relevance in this respect. These mutations do not only affect the infectivity and transmissibility of SARS-CoV-2, but also reduce the potency of vaccines, convalescent sera, and monoclonal antibody therapeutics (Harvey *et al*, 2021; Planas *et al*, 2021; Garcia-Beltran *et al*, 2021; Cele *et al*, 2021; Greaney *et al*, 2021; Lopez Bernal *et al*, 2021; Jangra *et al*, 2021; Takashita *et al*, 2022; Zhou *et al*, 2022; Peng *et al*, 2022). The recent emergence of the Delta and Omicron variants and symptomatic and sometimes even severe infections of doubly vaccinated people (Bergwerk *et al*, 2021; Farinholt *et al*, 2021; Christensen *et al*, 2022; Kuhlmann *et al*, 2022) is one such example, and more such variants will possibly develop. It remains to be determined whether increased natural immunity, mass vaccinations, and the increased application of therapeutic monoclonal antibodies and antiviral therapeutics will affect or drive further viral evolution. To prevent further severe disruptions to life and economies due to SARS-CoV-2, it is therefore paramount to design universal strategies for the prevention and treatment of current VOCs and possibly even to variants that will emerge in future.

Here, we report that soluble ACE2 (APN01), already being tested in clinical trials (NCT04335136), binds receptor-binding domain (RBD) and full-length Spike proteins of SARS-CoV-2 variants and especially VOCs with increased affinity when compared to the SARS-CoV-2 reference strain Spike and effectively neutralizes infections of all tested variants. Clinical Phase I testing of an inhalation approach, ultimately aiming at directly neutralizing SARS-CoV-2 at its site of entry, is currently underway (NCT05065645). Since Spike/ACE2 interaction is the crucial first step of viral infection, the viral Spike cannot mutate to escape ACE2 binding, without a loss in infectivity and tissue tropism. Our data provide the blueprint for a universal anti-COVID-19 agent with the potential to treat or even prevent infections against all current and potentially also future SARS-CoV-2 variants, as well as novel emerging coronaviruses using ACE2 as cell entry receptor.

# Results

### Enhanced affinity of clinical grade ACE2 to the Spike RBDs of emerging SARS-CoV-2 variants

Viral evolution of SARS-CoV-2 has been demonstrated to be focused on the Spike protein, which is instrumental for the early steps of viral infection (Pereson *et al*, 2021; Rochman *et al*, 2021). Many single or compound mutations, especially in the RBD of the viral Spike, have been described and either hypothesized or demonstrated to affect binding to the cell entry receptor ACE2 (see Harvey *et al*, 2021 for a review). To systematically test whether these emerged variants affect Spike/ACE2 interactions, we selected viral variants that have been described in the literature and in various databases. The RBD variants analyzed in this study and the location(s) of the respective mutations are depicted in a 3D model of the viral Spike RBD (Fig 1A). In an initial set of experiments, we performed ELISA analyses with plate-bound ACE2/APN01 to evaluate the impact of the indicated single and compound mutations on Spike RBD/ACE2 interactions. Intriguingly, almost all tested variant RBDs exhibited increased binding to APN01 (Appendix Fig 1A). These results prompted us to extend the number of analyzed variants to include current VOCs and to biophysically characterize RBD/ACE2 interactions using Biacore surface plasmon resonance (SPR) analysis.

For comparative kinetic binding analysis of SARS-CoV-2 RBD variants, dimeric APN01 was covalently coupled as ligand to optical sensor chips. Commercially available or in-house purified RBDs containing the amino acid changes described in Fig 1A were passed as analytes over the immobilized APN01 ligand in twofold serial dilutions. These proteins contained previously identified RBD mutations of current VOCs, as well as amino acid substitutions identified in variants of interest (VOI). Binding on-rates (association constants; $k_a$), off-rates (dissociation constants; $k_d$), and binding affinities ($K_D$ reported as nM) of SARS-CoV-2 RBD/ACE2 interactions were determined by mathematical sensorgram fitting, applying a monomeric Langmuir 1:1 interaction model $(A + B = AB)$ using BiaEvaluation 4.1 software. The results are summarized in Fig 1B, listing both the tested variants and the introduced amino acid substitution as well as the designation of the respective VOI and VOC that were tested with SPR analysis. Of note, the reference RBD sequence corresponds to the original Wuhan SARS-CoV-2 isolate. Importantly, our SPR analyses showed that affinities of all tested VOC RBDs to APN01 are substantially increased, with the Alpha variant RBD showing the highest affinity (Fig 1B and C); see Appendix Fig 1B for SPR sensorgrams of variants Kappa and Delta+). Of note, among these VOC RBDs, we observed a significantly lower off-rate for the Alpha variant and a significantly faster on-rate for the Delta variant. Binding of the Omicron variant was characterized by a high association rate, comparable to the Delta variant, but an off-rate reduction, as observed for the early variants of concern.

Combined with the increased binding affinity, changes in these kinetic parameters might contribute along with conformational changes of the whole Spike and the impact of mutations affecting proteolytic processing to the enhanced infectivity of VOC. These data show that, as expected and also in part reported by others (Cai *et al*, 2021; Gobeil *et al*, 2021; Kim *et al*, 2021; Ramanathan *et al*, 2021; Tchesnokova *et al*, 2021), viral evolution of the Spike protein led to an increase in Spike RBD/ACE2 binding affinity as well as altered kinetic constants of RBD/APN01 interaction, especially observed for VOC.

### Enhanced binding of SARS-CoV-2 VOC Spike trimers to clinical grade ACE2/APN01

SARS-CoV-2 RBD binding to ACE2 occurs in the context of trimeric Spike proteins in the pre-fusion conformation. Spike trimer

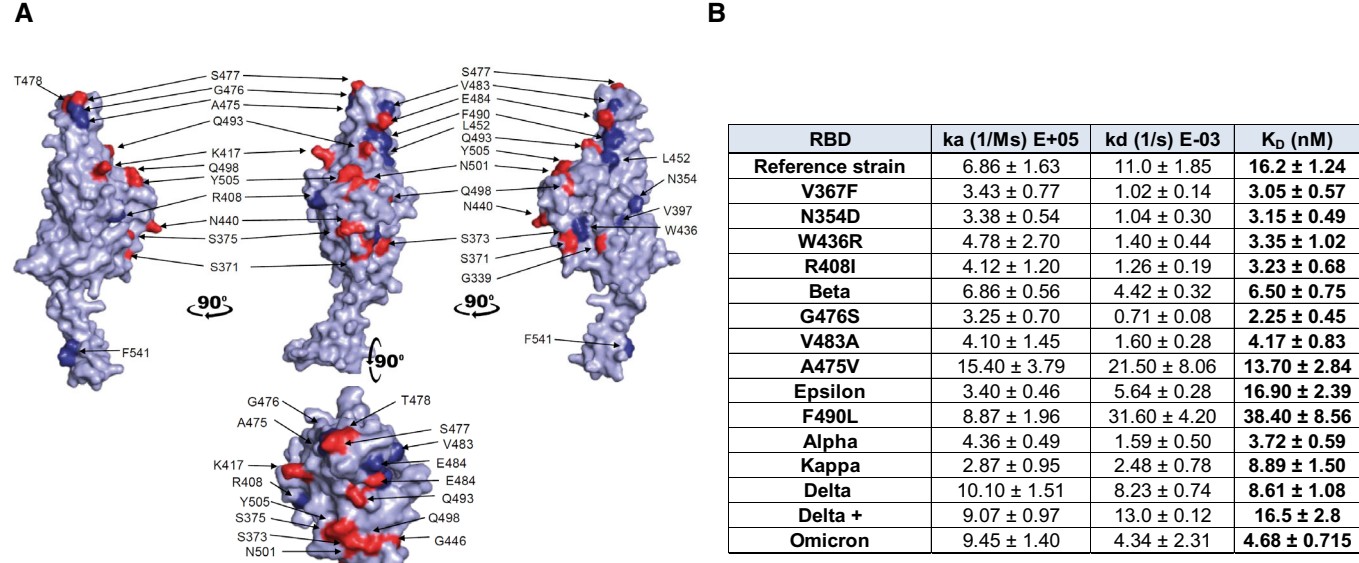

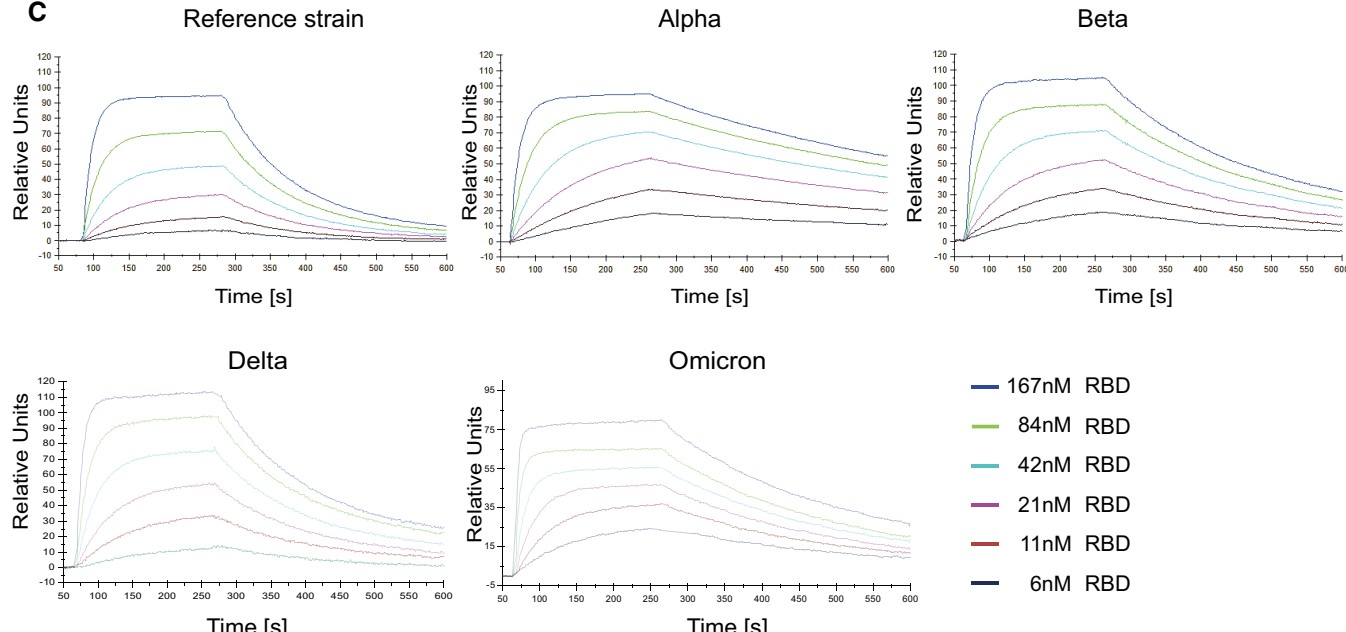

**Figure 1. Increased affinity of APN01 interactions with SARS-CoV-2-RBD variants.**

A  PyMOL rendered visualization of the SARS-CoV-2 RBD. Rendering depicts the SARS-CoV-2 RBD from the Wuhan reference isolate with mutation sites observed in SARS-CoV-2 variants shown in red. Indicated in blue are positions mutated in the various strains of SARS-CoV-2 used in experiments in this study.

B  Surface Plasmon Resonance analysis to derive kinetic constants ($k_a$, $k_d$) and affinity values ($K_D$) of SARS-CoV-2 RBD/APN01 interaction. The constants represent mean values and standard deviations obtained from sensorgram fittings performed in quadruplicate. The table lists both the tested variants and the introduced amino acid substitution as well as the designation of the respective Variant of Concern (VOC) or Variant of Interest (VOI) mutations tested in this study. Reference strain RBD sequence corresponds to the Wuhan SARS-CoV-2 isolate.

C  Representative SPR sensorgram images for the SARS-CoV-2 RBD/APN01 interaction of current VOCs.

structures have been solved in receptor bound or unbound forms (Walls *et al*, 2020; Wrapp *et al*, 2020; Cerutti *et al*, 2022; Mannar *et al*, 2022; Zhou *et al*, 2022), and it has been demonstrated that the conformation of the Spike (open or closed) is of critical relevance for receptor interactions (Cai *et al*, 2020; Xu *et al*, 2021).

Importantly, mutations outside the RBD that alter the Spike conformation have been reported to also affect viral infectivity and receptor binding (Mansbach *et al*, 2021; Yang *et al*, 2021). To test whether our observations on the increased affinity of VOC RBD/APN01 interaction is also observed in the context of the full-length

Spike, we assessed APN01 binding to recombinant pre-fusion trimeric SARS-CoV-2 Spike proteins. Since APN01 is a dimeric molecule thus allowing for bivalent target interaction, the VOC trimeric pre-fusion Spike variant proteins were immobilized to an optical sensor chip surface by covalent amine coupling. APN01 was passed over the immobilized Spike proteins in serial dilution in single binding cycles. Using BiaEvaluation 4.1 software, subsequent kinetic analysis was carried out by sensorgram fitting applying a Langmuir binding and a bivalent analyte binding model. Kinetic binding constants derived from the Langmuir model represent the apparent affinity. The bivalent analyte model calculates separate kinetic constants for the affinity determining first step A + B = AB and the avidity determining second step AB + B = $AB_2$ of the binding process.

Structural rendering of the trimeric full-length SARS-CoV-2 Spike protein and the positions mutated in the various strains of SARS-CoV-2 used in this study are shown in Fig 2A. Lineages and corresponding mutations of the SARS-CoV-2 isolates including VOCs and VOIs used in these biophysical analyses and cellular assays are listed in Fig 2B. SPR analysis showed strong binding of APN01 to all tested variants of the pre-fusion Spike trimers including the original Wuhan viral isolate trimer as well as the Alpha, Beta, Gamma, Delta, and Kappa, and Omicron variants (Fig 2C; Appendix Fig 2A). Sensorgram fitting showed enhanced apparent affinity (avidity; Langmuir model) and calculated first step affinity ($K_D1$, bivalent analyte model) of Alpha, Beta, Gamma, and Delta trimeric Spike proteins compared with Spike trimers of the reference strain (Fig 2C and D). For Omicron, no affinity enhancement was observed in this particular experimental setup. These results confirm that dimeric recombinant soluble human ACE2 (APN01) binds to the pre-fusion Spike trimers of all current VOCs with strong affinity/avidity.

### ACE2/APN01 effectively neutralizes SARS-CoV-2 VOCs

We have previously reported that clinical grade APN01 can effectively reduce the SARS-CoV-2 viral load in VeroE6 cells and 3D organoids in a dose-dependent manner, using a reference virus isolated early during the pandemic (Monteil et al, 2020). This virus carries the same Spike sequence as the originally reported virus isolated in Wuhan. To test whether APN01 can also neutralize variant clinical SARS-CoV-2 isolates including VOC strains (see Fig 3A for a list of tested strains), we performed neutralization assays in VeroE6 cells and VeroE6 cells overexpressing TMPRSS2 (VeroE6-TMPRSS2) and compared the inhibitory potency side by side to our reference strain. APN01 potently neutralized all the SARS-CoV-2 isolates we tested in VeroE6 cells (Fig 3B; Appendix Fig 3A). Intriguingly, this inhibition

was markedly enhanced against all the VOCs tested with sometimes up to 20 times lower $IC_{50}$ and $IC_{90}$ values (Fig 3C).

To extend our results to a physiologically more relevant cell system, we infected Calu-3 human lung epithelial cells with the reference SARS-CoV-2 isolate and the indicated SARS-CoV-2 variants. We again observed that clinical grade soluble APN01 potently reduced viral load of all tested variants in a dose-dependent manner (Fig 3C and D; Appendix Fig 3B). Importantly, the observed neutralization potency closely correlated with the Spike/APN01 binding affinity as assessed by SPR. In some cases, we detected 10 to 20 times lower $IC_{50}$ and $IC_{90}$ values when compared to values obtained with the SARS-CoV-2 reference strain. These data show that ACE2/APN01 not only binds significantly stronger to RBD or full-length Spike proteins of the tested variants, but also more potently inhibits viral infection by these strains in both VeroE6 and human lung epithelial cells.

### Independent validation studies

Experimental setups and model systems as well as culture conditions sometimes have a dramatic impact on experimental results. To ensure the reproducibility of our results, we conducted confirmatory experiments in a different and independent laboratory. These validation experiments were performed at the Karolinska Institutet, Stockholm, where VeroE6 cells were infected with reference virus isolated from the first Swedish patient. This virus was previously reported (Monteil et al, 2020, 2021) and carries the same Spike amino acid sequence as reported for the first Wuhan virus isolate. Experiments were performed at different Multiplicities of Infection (MOIs) to test the inhibitory potency of APN01. As reported before (Monteil et al, 2020), APN01 markedly reduced viral replication of the SARS-CoV-2 reference strain in a dose-dependent manner (Fig 4A and B). Importantly, the inhibitory potency of APN01 was again significantly increased toward the VOC Alpha and Beta, providing an independent validation of the neutralization results. This increase in neutralization potency of APN01, when the reference virus was compared with the VOC strains Alpha and Beta, could be observed at all APN01 concentrations and MOIs tested (Fig 4A–C).

In addition to altering the interaction with the cell entry receptor human ACE2, mutations in the SARS-CoV-2 Spike have been reported to affect potential transmission from humans to domesticated and wild animals, as has been reported for laboratory strains of mice (Hobbs & Reid, 2021; Kuchipudi et al, 2022) and https://www.cdc.gov/coronavirus/2019-ncov/daily-life-coping/animals.html). While this might have a positive impact on animal modeling of

---

**Figure 2. Increased binding affinity of APN01 to full-length pre-fusion trimeric Spike proteins from SARS-CoV-2 variants of concern.**

A  PyMOL rendering of the trimeric full-length SARS-CoV-2 Spike protein. One RBD is shown in violet. Indicated in blue are positions mutated in the various strains of SARS-CoV-2 used in experiments in this study. Mutations described for the Omicron VOC are depicted in red. Shown in yellow are the glycan modifications of the spike protein (Capraz et al, 2021).

B  Table lists the SARS-CoV-2 strains and their respective mutations within the Spike protein that were used in this study.

C  Representative sensorgram images for the SPR analysis conducted with full-length trimeric spike proteins in pre-fusion state with APN01. Reference strain corresponds to original Wuhan viral isolate spike sequence. Indicated are VOC Alpha, Beta, Gamma, and Delta, and Omicron.

D  Tables listing $k_a$, $k_d$, as well as $K_D$ values for the interaction of APN01 and full-length trimeric spike proteins. The constants represent mean values and standard deviations obtained from sensorgram fittings performed in quadruplicate. Values are derived from calculations based upon the Langmuir (left table) or Bivalent Analyte sensorgram fitting (right table).

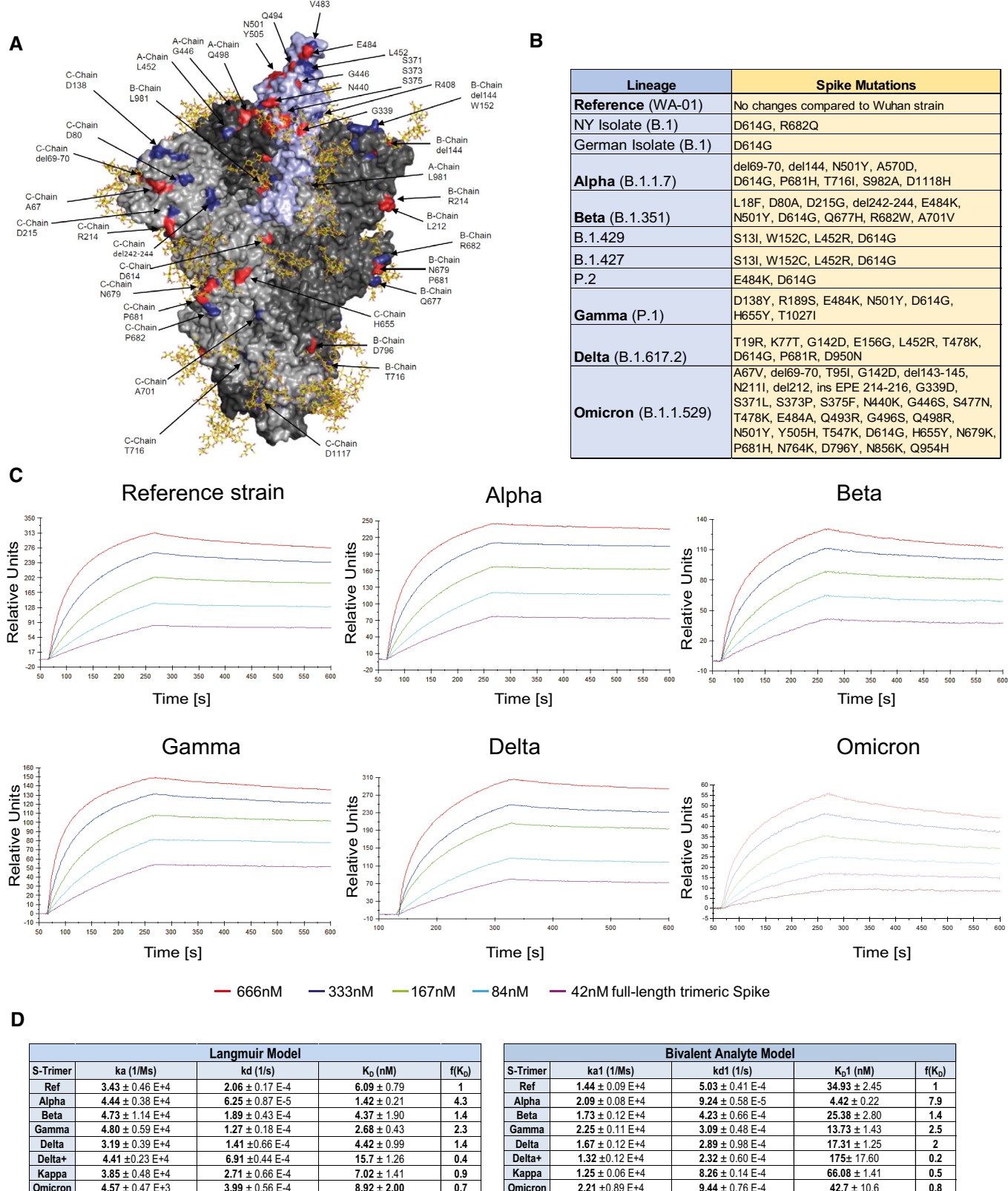

**Figure 2.**

SARS-CoV-2 infection, this is also of potentially high concern due to the possibility of additional viral reservoirs, as well as the accumulation of mutations that might affect viral fitness upon transmission back to human hosts. We indeed observed that SARS-CoV-2 reference strain infection of Calu-3 cells could not be inhibited by recombinant mouse ACE2 but the Omicron infection was potently

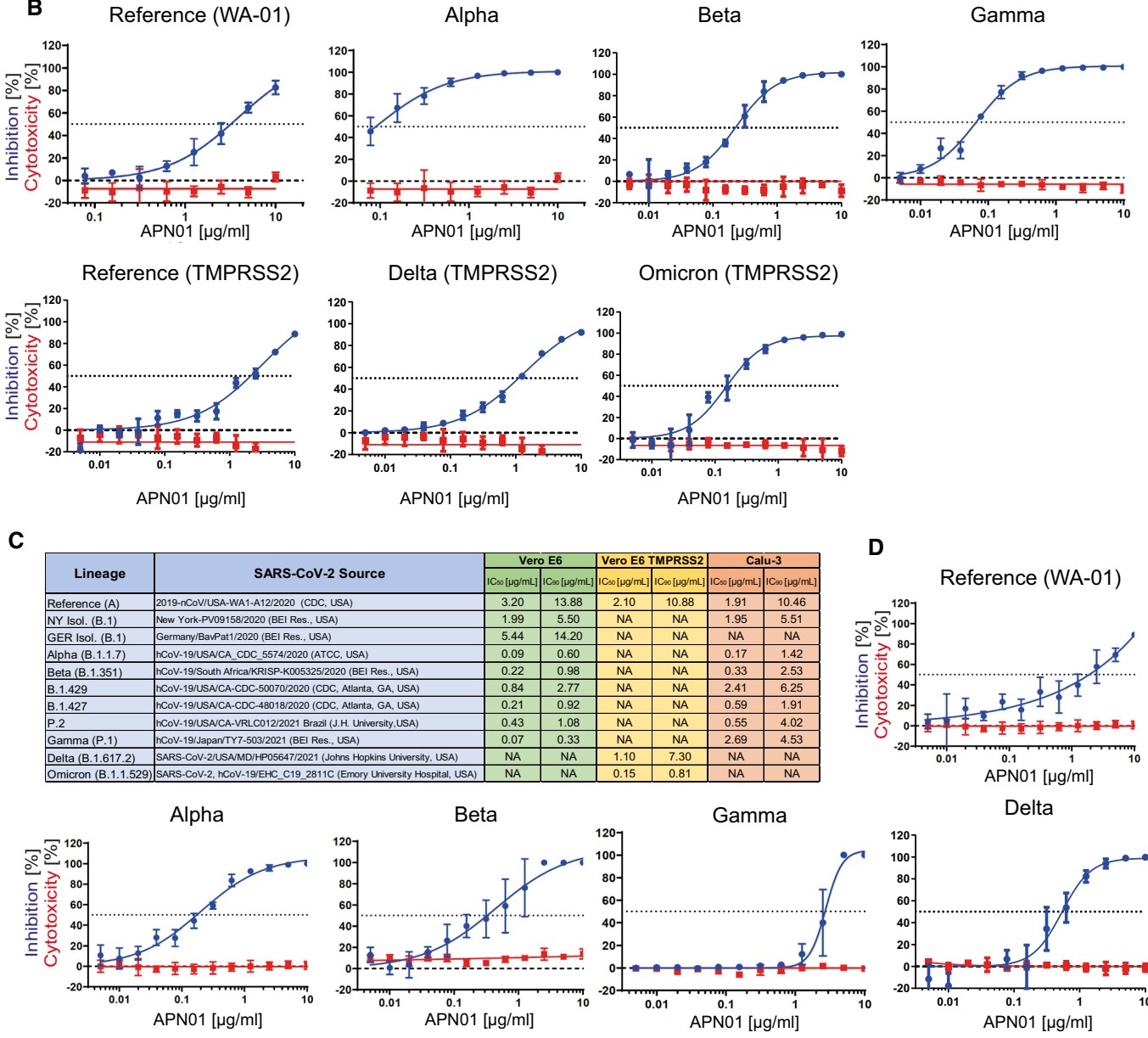

**Figure 3.**

**Figure 3.  Increased neutralization potency of APN01 toward SARS-CoV-2 variants.**

A   Table depicts source of the tested viral isolates, as well plaque-forming units (PFU) and the infection time used in these experiments for both VeroE6/VeroE6-TMPRSS2 and Calu-3 cells.

B   Panels depict both neutralization of the indicated SARS-CoV-2 isolates (blue line) and cytotoxicity of APN01 (red line) in VeroE6 or VeroE6-TMPRSS2 cells. Analysis was done in quadruplicate with mean and standard deviations shown. Y-axis depicts the percentage of neutralization and cytotoxicity, respectively.

C   Table depicts $IC_{50}$ and $IC_{90}$ values for APN01-mediated neutralization of viral infection in VeroE6/VeroE6-TMPRSS2 and Calu-3 cells.

D   Same experimental setup as in (a) but conducted with the epithelial lung cancer cell line Calu-3. Analysis was done in quadruplicate with mean and standard deviations shown. Y-axis depicts the percentage of neutralization and cytotoxicity, respectively.

suppressed by both human and mouse recombinant ACE2 for the indicated MOIs, albeit at lower potency when using soluble mouse ACE2 (Appendix Fig 4). These data are in line with reported findings on the effect of some VOC Spike mutations on transmissibility to non-human hosts.

While the origin of the Omicron VOC is still not fully clarified, the high number of mutations in Omicron has raised concerns in terms of immune evasion and the protective effect of currently available vaccines and therapeutic antibodies. To test the potential impact of these mutations on the protection status of vaccinated individuals, we compared the potency of vaccinees' sera to neutralize both reference (Wuhan) and the Omicron SARS-CoV-2 virus. In support of previous observations, vaccinees' sera exhibited a significantly decreased potency in viral neutralization when reference and Omicron infections of VeroE6 cells were compared (Appendix Fig 5). Our data are in line with recently published observations on the waning efficacy of current vaccines, as well as a (partial) loss of neutralization potency of many currently marketed or developed monoclonal antibodies (Cele *et al*, 2021; Garcia-Beltran *et al*, 2021; Greaney *et al*, 2021; Harvey *et al*, 2021; Jangra *et al*, 2021; Lopez Bernal *et al*, 2021; Planas *et al*, 2021; Peng *et al*, 2022; Takashita *et al*, 2022; Zhou *et al*, 2022). The results described in this study, using different viral isolates as well as different experimental procedures, independently confirm that clinical grade soluble human APN01 strongly blocks SARS-CoV-2 infections of recently emerged VOC and indicate that this therapeutic approach provides a potentially universally efficacious therapeutic approach to treat all current and future variants of SARS-CoV-2.

# Discussion

Since emerging from Wuhan, China, in December of 2019, SARS-CoV-2 has been causing devastating severe respiratory infections in humans worldwide. Multiple variants of SARS-CoV-2 have emerged and circulated around the world throughout the COVID-19 pandemic with some strains displaying even greater infectivity and transmissibility (see https://www.cdc.gov/coronavirus/2019-ncov/variants/variant-info.html and https://www.who.int/en/activities/tracking-SARS-CoV-2-variants/ for further information). Some of these variants have been classified by the WHO as *Variants of Concern* (VOC), defined as "a variant for which there is evidence of an increase in transmissibility, more severe disease, significant reduction in neutralization by antibodies generated during previous infection or vaccination, reduced effectiveness of treatments or vaccines, or diagnostic detection failures" or *Variants of Interest* (VOI), defined as "a variant with specific genetic markers that have been associated with changes to receptor binding, reduced

neutralization by antibodies generated against previous infection or vaccination, reduced efficacy of treatments, potential diagnostic impact, or predicted increase in transmissibility or disease severity." The current VOCs are B.1.1.7 (Alpha), B.1.351 (Beta), P.1 (Gamma), B.1.617.2 (Delta), and in particular Omicron (B.1.529), whereas multiple VOI are currently circulating as well. Both large-scale vaccination programs, emerging treatments as well as protracted infections in immune-compromised hosts or animal reservoirs, could potentially drive further molecular evolution of SARS-CoV-2. It is, therefore, to be expected that novel variants will arise, some of which will rapidly spread and even re-infect fully vaccinated people, sometimes leading to severe breakthrough COVID-19, as has been observed for the Delta VOC and is currently observed for the Omicron variant (Farinholt *et al*, 2021; Christensen *et al*, 2022). In this arena of global vaccination efforts and antibody therapeutics in multiple permutations, it therefore remains critical to identify and design universal strategies that might help prevent and treat infections with all current and potential future variants.

The SARS-CoV-2 Spike protein interacts with high affinity with its main entry receptor ACE2, followed by a subsequent membrane fusion step. Although alternative ACE2-independent modes of viral uptake and infection have been reported, genetic modeling in ACE2 deficient mice has provided unambiguous evidence that ACE2 is the essential receptor for SARS-CoV-2 infections and subsequent development of COVID-19 in mice (Gawish *et al*, 2022). Indeed, most neutralizing antibodies from vaccinations, convalescent plasma therapies, and monoclonal antibodies or nanobodies interfere with the Spike/ACE2 interaction (Harvey *et al*, 2021) and have, due to their therapeutic or prophylactic efficacy, provided further evidence for the essential role of ACE2 as the relevant *in vivo* receptor for SARS-CoV-2 and COVID-19 in humans. Conceptually, all SARS-CoV-2 variants and "escape mutants" still bind to ACE2 (Cai *et al*, 2021; Gobeil *et al*, 2021; Motozono *et al*, 2021; Ou *et al*, 2021; Tchesnokova *et al*, 2021; Tian *et al*, 2021; Yuan *et al*, 2021; Zhou *et al*, 2021), in part already validated in various studies (Cai *et al*, 2021; Yuan *et al*, 2021; Zhang *et al*, 2021). ACE2 is a carboxypeptidase that controls angiotensin II peptide levels and thereby is involved in critical aspects of physiology such as blood pressure control or sodium retention as well as disease processes including heart failure, blood vessel and kidney pathologies in diabetes, tissue fibrosis, or regulation of inflammatory cytokines (Kuba *et al*, 2010). Our group was the first to show that soluble dimeric ACE2 protects mice from acute lung injury and acute respiratory distress syndrome (ARDS) (Imai *et al*, 2005), which triggered preclinical and clinical development of recombinant human soluble ACE2 (Treml *et al*, 2010; Haschke *et al*, 2013), termed APN01, for lung disease. It was, therefore, critical to systematically assess

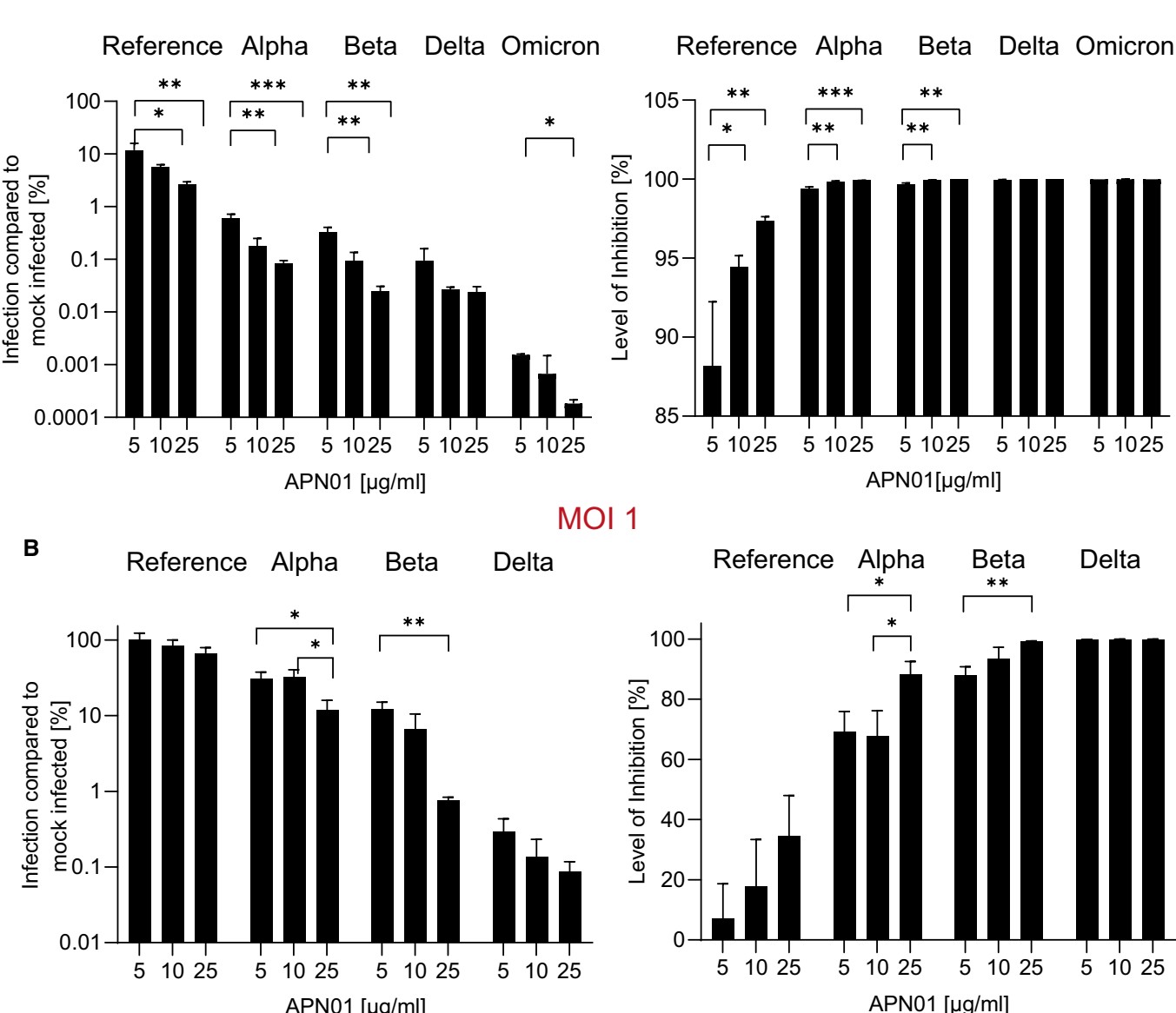

## C

| Lineage | Virus Source and Isolate | Vero E6 MOI 0.01 | | Vero E6 MOI 1 | |
|---|---|---|---|---|---|
| | | IC$_{50}$ (µg/ml) | IC$_{90}$ (µg/ml) | IC$_{50}$ (µg/ml) | IC$_{90}$ (µg/ml) |
| Reference | hCoV-19/Sweden/01/2020 | 0.7705 | 5.904 | 44.49 | 337.092 |
| Alpha | hCOV-19/Sweden/21-53846/2021 | 0.2062 | 0.812 | 1.747 | 58.959 |
| Beta | hCOV-19/Sweden/21-51217/2021 | 0.2007 | 0.69 | 1.028 | 6.099 |
| Delta | SARS-CoV-2/hu/DK/SSI-H1 | 0.02974 | 0.149 | 0.00712 | 0.0836 |
| Omicron | hCOV-19/Sweden/21-55629/2021 | 0.00063 | 0.00374 | - | - |

**Figure 4. Increased potency of APN01 against SARS-CoV-2 VOC.**

A, B  Diagrams depict the level of infection with the indicated SARS-CoV-2 isolates at MOI 0.01 (A) and MOI 1 (B) of VeroE6 cells after pre-incubation with increasing concentrations of APN01 as compared to infections in the absence of APN01 pre-incubation. MOIs as depicted reflect *multiplicities of infection* before the pre-incubation with APN01 that was followed by the viral infection of cells. Shown are means of biological replicates (*n* = 3) analyses with standard deviations. Statistical significance is indicated by asterisks (*$P$-value < 0.05; **$P$-value < 0.01, ***$P$-value < 0.001 as calculated with one-way ANOVA).

C  List and source of strains used at the Karolinska Institutet and IC$_{50}$ and IC$_{90}$ values obtained for the indicated MOIs. See Materials and Methods section for a detailed list of viral mutations for the strains used.

whether APN01 could indeed bind to the RBD and more importantly full-length Spike of emerged SARS-CoV-2 variants and neutralize infections with these variants—especially recently emerged VOC. Our data show that APN01 associates with the RBD and pre-fusion trimeric Spike of all variants tested, often at markedly increased affinity and avidity, which is in line with the enhanced infectivity of these variants. Importantly, using different cell types including lung epithelial cells, APN01 showed markedly improved efficacy to block infection of all VOC, demonstrating that the prediction holds true—clinical grade APN01 can effectively block all tested SARS-CoV-2 variants and this inhibition is markedly improved against all tested VOCs and VOIs.

APN01 has now undergone phase 2 testing in severe (WHO scores 4, 5, and 6) COVID-19 patients using intravenous infusions. Additionally, a new formulation of APN01 that can be inhaled as an aerosol to directly interfere with the earliest steps of viral infection and prevent lung damage and spreading to other organ systems has been developed (preprint: Shoemaker *et al*, 2021). Moreover, the preclinical efficacy of APN01 lung administration in a mouse-adapted severe COVID-19 model has been demonstrated, providing proof of concept for phase 1 clinical inhalation trials in humans that are currently ongoing. Of note, the FDA recently expanded emergency use authorization for a two-antibody "cocktail" to allow its use in patients seeking protection from COVID-19 following exposure to someone infected with SARS-CoV-2, providing a framework for preventive and post-exposure prophylactic strategies. However, structural and functional analyses of the VOCs reveal decreased efficacy and viral escape from therapeutic antibodies or antibodies generated from natural infection and vaccinations (Zhou *et al*, 2022). Further viral evolution, especially under selective pressure by both widespread vaccination programs and monoclonal antibodies or antibody cocktails, might drive the emergence of resistant novel variants and strains. In particular, the Delta and Omicron VOC strains have already substantially contributed to a new global wave of infection and concerningly large numbers of re-infections. Our data show that clinical grade soluble ACE2/APN01 blocks infectivity of all tested variants. As expected from the stoichiometry of these interactions, this neutralization was dependent upon the ratio of infectious particles to neutralizing APN01, that is, APN01 potency correlated with the number of infectious particles. These results support the notion that this therapeutic is inherently resistant to escape mutations that constitute a major problem for both vaccines and antibody-based therapeutics. Data provided in this study and in (preprint: Shoemaker *et al*, 2021) as well as clinical data already generated for APN01 set the stage for the development of universal and pan-variant SARS-CoV-2 prevention and therapy.

### Limitations of this study

Our study used two different cell types and should be expanded using additional human cell types and also organoids. Additionally, different incubation times with viral nucleoprotein staining as experimental readout might affect the accuracy of the obtained experimental results. Moreover, to indeed make a claim on universality, additional variants should (and can) be tested using affinity/avidity measurements and neutralization assays.

## Materials and Methods

### Surface plasmon resonance analysis

Recombinant SARS-CoV-2 spike proteins (His-tagged) were purchased from Acro Biosystems Inc. (Newark, USA) or produced in-house by overexpression of respective constructs in HEK293T cells followed by purification *via* nickel NTA chromatography and size exclusion chromatography. The purity of recombinant proteins was documented by SDS–PAGE analysis. SPR measurements were performed on a Biacore 3000 instrument (GE Healthcare). For comparative kinetic analysis of SARS-CoV-2 RBD variants, APN01, which does not contain a capture tag, was immobilized on optical sensor chip surfaces by covalent coupling at pH = 4.5 at ligand densities of $2,672 \pm 145$ RU following the Biacore amine coupling protocol. Amine-activated flow cell 1 (FC1) was used as a reference to allow for the generation of background-subtracted binding sensorgrams. SARS-CoV-2 RBD protein variants were passed over the immobilized APN01 ligand as analytes in twofold serial dilution (167, 84, 42, 21, 11, and 6 nM) in single binding cycles, at a flow rate of 30 μl/min in HBS-EP buffer (0.1 M HEPES, 1.5 M NaCl, 0.03 M EDTA and 0.5% v/v Surfactant P20). Bound analyte was removed after each cycle by surface regeneration with 3 M $MgCl_2$. Reference flow cell (FC1) subtracted sensorgram overlays with additional correction by subtracting buffer (c = 0) sensorgrams (double referencing) were generated and used for kinetic binding analysis. Subtraction spikes occurring at the injection start were removed in the sensorgrams shown in the figures. Kinetic binding constants ($k_a$, $k_d$, and $K_D$) were generated by mathematical sensorgram fitting. Generally, a Langmuir 1:1 interaction model (A + B = AB) was applied, using BiaEvaluation 4.1 software. A series of 4 curve fittings was performed for each binding reaction, using a simultaneous single fitting algorithm for each of the sensorgram overlays. Mean values and standard deviations were obtained from fitting runs with $Chi^2$ values $\leq 3\%$ of $R_{max}$. Binding affinities (reported as nM) were calculated from on- and off-rate constants.

SPR analysis of APN01 binding to recombinant full-length trimeric SARS-CoV-2 spike proteins was performed with the spike proteins immobilized to CM5 optical sensor chip surfaces by covalent amine coupling to reach a surface density of 900–1,000 RU. APN01 was passed over the immobilized spike proteins as dimeric bivalent analyte at five concentrations (42, 84, 167, 333 and 666 nM) in repetitive single binding cycles. Using BiaEvaluation 4.1 software, kinetic analysis was carried out by applying a Langmuir 1:1 (apparent affinity) and a bivalent analyte binding algorithm, which separates first step binding (A + B = AB; affinity) from the second step binding ($AB + B = AB_2$; avidity).

### Cell lines and cell culture

Infection and APN01-mediated viral neutralization assays were conducted at the Integrated Research Facility at Fort Detrick (IRF-Frederick) of the National Institute of Allergy and Infectious Diseases (NIAID) or at the Department of Laboratory Medicine (Unit of Clinical Microbiology) of the Karolinska Institutet and Karolinska University Hospital. Vero E6 cells (Vero C1008; American Type Culture Collection [ATCC], Manassas, VA, USA) were cultured in DMEM medium (Gibco, Gaithersburg, MD, USA or Thermo Fisher)

containing 10% fetal bovine serum (FBS). Calu-3 cells (HTB-55; American Type Culture Collection [ATCC], Manassas, VA, USA) were cultured in DMEM F12 Medium (ATCC) with 20% FBS. Both cell lines were tested for mycoplasma.

### Viral neutralization experiments conducted at the Karolinska Institutet

A 24 h after seeding of Vero E6 cells ($5 \times 10^4$ per 48 well), APN01 was mixed with viral particles of the indicated strains at the given concentrations in DMEM Medium (Thermofisher) containing 5% FBS in 100 µl per well and incubated for 30 min at 37°C. After the incubation period medium was removed from Vero E6 cells, cells were washed once with PBS to remove any non-attached cells and virus/APN01 mixtures were added to the cells. Cells were incubated with virus for 15 h, after which cells were washed three times with PBS and lysed the Trizol, subsequently. RNA was extracted using the direct-zol RNA kit (Zymo Research) and assayed by qRT–PCR as previously described (Monteil et al, 2020). Half-maximal inhibitory concentration ($IC_{50}$) and inhibitory concentration 90 ($IC_{90}$) were calculated using GraphPad Prism Software (La Jolla, CA). For serum neutralization assays, Vero E6 cells were seeded in 48-well plates as described above. 24-h post-seeding, indicated dilutions of vaccinated subjects sera were mixed with SARS-CoV-2 Wuhan or Omicron strains at an MOI of 0.01 in a final volume of 100 ml per well in DMEM (5% FBS) at 37°C under shaking conditions for 30 min. The serum dilutions used in this experiment were determined after a neutralization assay against the Wuhan reference strain. After 30 min, Vero E6 cells were infected with Serum/301 SARS-CoV-2 for 15 h. A 15-h post-infection, supernatants were removed, and cells were washed 3 times with PBS and then lysed using Trizol (Thermofisher) before analysis by qRT–PCR for viral RNA detection as previously described (Monteil et al, 2020).

### Viral neutralization experiments conducted at the IRF-Fredrick

For infection experiments, Vero E6 cells were seeded at 6,000 cells in 30 µl in 384 wells 24 h prior to infection and Calu-3 cells were plated at 10,000 cells per well in 30 µl 48 h prior to infection in their respective culture media. For APN01 neutralization experiments, APN01 was diluted twofold in an eight- or twelve-point dose curve. Each condition was tested in quadruplicate ($n = 4$) with an efficacy plate and a mock-infected cytotoxicity plate run in parallel. Strains for infection and MOIs are listed in a table in the respective figures. Suitable MOIs were optimized previously for each cell line and virus using serial dilutions and staining for SARS-CoV-2 nucleoprotein, as described below. The viral inoculum was diluted in the respective cell culture medium containing indicated doses of APN01 and pre-incubated for 60 min. 20 µl of the virus/APN01 mixture was transferred directly to plates containing cells to reach a final volume of 50 µl. After infection, cells were incubated for 24 h or 48 h, in accordance with optimal virus and cell-line infection conditions. Cells were then fixed in 10% formalin and stained with a SARS-CoV-1 nucleoprotein-specific antibody (cross-reactive to SARS-CoV-2; see antibody list for further information), followed by a secondary antibody conjugated with fluorophore and/or horseradish peroxidase (HRP). Fluorescence was quantitated using a PerkinElmer Operetta high-content imaging system (PerkinElmer, Massachusetts,

USA). Chemiluminescence was read on a Tecan M1000 plate reader (Tecan, Switzerland). Cytotoxicity on mock-infected plates was determined using the Promega CellTiter-Glow Luminescent Cell Viability Assay (Promega, Madison, Wisconsin, USA). Half-maximal inhibitory concentration ($IC_{50}$) and 50% cytotoxic concentration ($CC_{50}$) were calculated as described by Covés-Datson et al (2019), using GraphPad Prism Software (La Jolla, CA). Z' factor scores were assessed as quality-control parameters for each plate of each run. All plates included in the report passed quality-control criteria.

### Viral strains and isolates

Sources and strains used in the experiments at the NIAID and Karolinska Institutet are indicated in the respective figures, including a list of SARS-CoV-2 spike mutations identified by sequencing of viral isolates (NIAID). SARS-CoV-2 variant Delta was provided by Charlotta Polacek Strandh, Statens Serum Institut, Copenhagen, Denmark. Experiments dealing with SARS-CoV-2 were performed in a BSL-3 laboratory under the approval of Public Health Agency of Sweden. Mutations of viral isolates used at Karolinska are listed in Appendix Table 1.

### Preparation of recombinant human ACE2

Clinical grade recombinant human ACE2 (amino acids 18–740) was produced by the contract manufacturer Polymun Scientific (Klosterneuburg, Austria) from CHO cells according to GMP guidelines under serum-free conditions and formulated as a physiologic aqueous solution, as described previously (Haschke et al, 2013; Zoufaly et al, 2020).

### ELISA experiments

96-well ELISA plates were coated with 100 µl of anti-ACE2 coating antibody (2 µg/ml diluted in PBS pH 7.4) overnight at room temperature. Following coating, plates were washed 3 times with 300 µl of washing buffer (PBS + 0.05% Tween-20) and blocked with 300 µl of blocking buffer (1% BSA in PBS, pH 7.4) for 1 h at room temperature. After blocking, plates were washed five times and 100 µl APN01, diluted in blocking buffer to a concentration of 2 µg/ml, was applied to the wells and incubated for 1 h at room temperature. Subsequently, plates were washed five times with washing buffer and 100 µl of SARS-CoV-2 Spike RBD variants were added to the plates in triplicates. As controls for non-specific binding, wells were incubated in washing buffer without RBD proteins. Following incubation for 1 h at room temperature, plates were washed five times with washing buffer after which 100 µl of anti-Histidine detection antibody (diluted 1/500 in blocking buffer) was added for another hour. After five washing steps, 100 µl secondary HRP conjugated antibody was added for 1 h at room temperature. Subsequently, the plates were washed seven times with washing buffer and 100 µl substrate (TMB Microwell Peroxidase Substrate, Seramun Diagnostika #S-001-2-TMB, ready-to-use) was added to the wells. The colorimetric reaction was stopped by the addition of 50 µl of 1 M sulfuric acid and subsequently analyzed for absorption at 450 nm. To measure unspecific background binding, control values were generated by omitting the addition of RBD proteins. These control values were subtracted from RBD-specific signals and plotted as OD values.

**The paper explained**

**Problem**

SARS-CoV-2 infections caused an unprecedented global pandemic with millions of fatalities and morbidities. While great progress has been made in vaccinations, drugs for the treatment of COVID-19, including antibodies, convalescent sera, and even small molecule-based therapeutics, are inherently vulnerable to viral escape mutations. Consequently, there is a large unmet medical need for therapeutics that can act on all current and future SARS-CoV-2 variants to limit the progression of COVID-19.

**Results**

In this study, we provide evidence for the therapeutic potential of soluble recombinant human ACE2 as a pan-SARS-CoV-2 molecular decoy therapeutic approach. We systematically tested the binding of SARS-CoV-2 variants with a focus on variants of concern to recombinant soluble human ACE2 and provide evidence for an increase in binding affinity and avidity for all tested variants of concern. This increased binding affinity correlates with a substantially enhanced potency of the soluble ACE2 decoy toward all variants of concern, including Delta and Omicron, when compared to the original viral isolate of SARS-CoV-2. Data from an independent laboratory confirm these findings.

**Impact**

The efficacy of soluble ACE2 toward all tested variants of concern, including the delta and omicron sub-lineages, highlights the therapeutic potential of ACE2 a universal strategy to treat potentially all current and future SARS-CoV-2 variants and additional coronaviruses that use ACE2 as a cell entry receptor. Clinical studies on the safety and tolerability of soluble ACE2 inhalation are currently ongoing to provide the basis to test its efficacy to prevent and treat SARS-CoV-2 infections, an approach that should be feasible for a large population.

**Visualizations of RBDs and full-length Spike protein**

Visualizations were rendered with pymol software (the PyMOL Molecular Graphics System, Version 2.4 Schrödinger, LLC), based on a model of the fully glycosylated Spike-ACE2 complex described in Capraz *et al* (submitted for publication) and https://covid.molssi.org//models/#spike-protein-in-complex-with-human-ace2-ace2-spike-binding.

**Sera of vaccinees**

Sera were taken 5–7 weeks after the second immunization with the mRNA vaccine Comirnaty (median dose interval 21 days (range 21–24) from four SARS-CoV-2-naïve healthcare workers (75% female, median age 46 [IQR 37–59]) which took part in the COMMUNITY study. The COMMUNITY study was approved by the Swedish Ethical Review Authority (Dnr: 2020–01653). Informed consent was obtained from all subjects and experiments conformed to the principles set out in the WMA Declaration of Helsinki and the Department of Health and Human Services Belmont Report.

Primers and antibodies used in this study are listed in Appendix Tables 2 and 3.

# Data availability

This study includes no data deposited in external repositories.

Expanded View for this article is available online.

## Acknowledgements

We like to thank employees of invIOs' R&D department for input and critical discussion during the course of this study. We would also like to acknowledge Mr. Gregory Kocher (IRF-NIAID-NIH) for growing SARS-CoV-2 virus variants. J.M.P. and A.H. and the research leading to these results has received funding from the T. von Zastrow foundation, the FWF Wittgenstein award (Z 271-B19), the Austrian Academy of Sciences, the Innovative Medicines Initiative 2 Joint Undertaking (JU) under grant agreement No 101005026, and the Canada 150 Research Chairs Program F18-01336 as well as the Canadian Institutes of Health Research COVID-19 grants F20–02343 and F20-02015. Additionally, this project has received funding from the Innovative Medicines Initiative 2 Joint Undertaking (JU) under grant agreement no. 101005026. The JU receives support from the European Union's Horizon 2020 research and innovation program and EFPIA. Parts of this project have been funded with federal funds from the National Cancer Institute, National Institutes of Health, under Contract No. 75N91019D00024, Task Order No. 75N91019F00130. O.H.A. has received a grant from the Swiss National Science Foundation (P400PM_194473). Parts of this project have been funded with federal funds from the National Cancer Institute, National Institutes of Health, under Contract No. 75N91019D00024, Task Order No. 75N91019F00130 and through Laulima Government Solutions, LLC prime contract with the US National Institute of Allergy and Infectious Diseases (NIAID) under Contract No. HHSN272201800013C (B.E., M.R.H.). M.M. and E.P. performed this work as employees of Tunnell Government Services (TGS), a subcontractor of Laulima Government Solutions, LLC. The content of this publication does not necessarily reflect the views or policies of the Department of Health and Human Services, nor does mention of trade names, commercial products, or organizations imply endorsement by the U.S. Government (IC).

## Author contributions

**Vanessa Monteil:** Conceptualization; data curation; formal analysis; visualization; methodology. **Brett Eaton:** Data curation; formal analysis. **Elena Postnikova:** Data curation; formal analysis; investigation. **Michael Murphy:** Conceptualization; data curation. **Benedict Braunsfeld:** Data curation; software; visualization. **Ian Crozier:** Conceptualization. **Franz Kricek:** Conceptualization; data curation; formal analysis. **Janine Niederhoefer:** Data curation; formal analysis; methodology. **Alice Schwarzboeck:** Investigation. **Helene Breid:** Data curation. **Stephanie Devignot:** Formal analysis. **Jonas Klingström:** Formal analysis. **Charlotte Thalin:** Formal analysis. **Max J Kellner:** Formal analysis. **Wanda Christ:** Formal analysis. **Sebastian Havervall:** Formal analysis. **Stefan Mereiter:** Conceptualization. **Sylvia Knapp:** Conceptualization. **Anna Sanchez Jimenez:** Methodology. **Agnes Bugajska-Schretter:** Conceptualization. **Alexander Dohnal:** Conceptualization. **Christine Ruf:** Formal analysis; investigation. **Romana Gugenberger:** Conceptualization. **Astrid Hagelkruys:** Conceptualization; funding acquisition. **Nuria Montserrat:** Conceptualization. **Ivona Kozieradzki:** Data curation. **Omar Hasan Ali:** Formal analysis. **Johannes Stadlmann:** Conceptualization. **Michael R Holbrook:** Conceptualization; data curation; formal analysis. **Connie Schmaljohn:** Conceptualization. **Chris Oostenbrink:** Conceptualization; software; formal analysis. **Robert H Shoemaker:** Conceptualization. **Ali Mirazimi:** Conceptualization; data curation; formal analysis. **Gerald Wirnsberger:** Conceptualization; data curation; formal analysis; supervision; funding acquisition; validation; investigation; visualization; methodology; project administration. **Josef M Penninger:** Conceptualization.

## Disclosure and competing interests statement

J.M.P. declares a conflict of interest as a founder and shareholder of Apeiron Biologics. G.W., R.G., A.S., J.N, A.S.J, and H.B. are employees of invIOs. Apeiron holds a patent on the use of ACE2 for the treatment of various diseases and is currently testing APN01 (soluble recombinant human ACE2) for the treatment of COVID-19. Josef Penninger is an EMBO Member. This has no bearing on the editorial consideration of this article for publication. The other authors declare that they have no conflict of interest.

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
