## [Review Process File · EMBO Molecular Medicine]

Clinical grade ACE2 as a universal agent to block SARS-CoV-2 variants

Vanessa Monteil, Brett Eaton, Elena Postnikova, Michael Murphy, Benedict Braunsfeld, Ian Crozier, Franz Kricsek, Janine Niederhoefer, Alice Schwarzboeck, Helene Breid, Stephanie Devignot, Jonas Klingström, Charlotte Thalín, Max Kellner, Wanda Christ, Sebastian Havervall, Stefan Mereiter, Sylvia Knapp, Anna Sanchez Jimenez, Agnes Bugajska-Schretter, Alexander Dohnal, Christine Ruf, Romana Gugenberger, Astrid Hagelkruys, Nuria Montserrat, Ivona Kozieradzki, Omar Ali, Johannes Stadlmann, Mike Holbrook, Connie Schmaljohn, Chris Oostenbrink, Robert Shoemaker, Ali Mirazimi, Gerald Wirnsberger, and Josef Penninger

DOI: 10.15252/emmm.202115230

Corresponding author(s): Josef Penninger (josef.penninger@imba.oeaw.ac.at) , Gerald Wirnsberger (gwi@invios.com)

Review Timeline:

Submission Date:	30th Sep 21
Editorial Decision:	3rd Nov 21
Revision Received:	12th Apr 22
Editorial Decision:	10th May 22
Revision Received:	3rd Jun 22
Accepted:	3rd Jun 22

Editor: Jingyi Hou

Transaction Report:

3rd Nov 2021

Dear Prof. Penninger,

Thank you for submitting your work to EMBO Molecular Medicine. We have now heard back from the three referees who evaluated your manuscript. As you will see from the reports below, the referees acknowledge the potential interest and relevance of the study. However, they also raise a series of concerns about your work, which should be convincingly addressed in a major revision of the present manuscript.

The referees' recommendations are rather straightforward, and there is no need to reiterate their comments. Importantly, the referees pointed out that different viral input and infection times were used for the different VOCs and mentioned that the delta variant was not tested in some cell culture-based assays, which need to be carefully addressed.

All other issues need to be addressed as well. We would welcome the submission of a revised version within three months for further consideration. Please note that EMBO Molecular Medicine strongly supports a single round of revision. As acceptance or rejection of the manuscript will depend on another round of review, your responses should be as complete as possible.

EMBO Molecular Medicine has a "scooping protection" policy, whereby similar findings that are published by others during review or revision are not a criterion for rejection. Should you decide to submit a revised version, I do ask that you get in touch after three months if you have not completed it to update us on the status.

We are aware that many laboratories cannot function at full efficiency during the current COVID-19/SARS-CoV-2 pandemic and have therefore extended our "scooping protection policy" to cover the period required for a full revision to address the experimental issues. Please let me know should you need additional time, and also if you see a paper with related content published elsewhere.

I look forward to seeing a revised form of your manuscript as soon as possible.

Use this link to login to the manuscript system and submit your revision: <https://embomolmed.msubmit.net/cgi-bin/main.plex>

Kind regards,
Jingyi

Jingyi Hou
Editor
EMBO Molecular Medicine

We require:

- 1) A .docx formatted version of the manuscript text (including legends for main figures, EV figures and tables). Please make sure that the changes are highlighted to be clearly visible.
- 2) Individual production quality figure files as .eps, .tif, .jpg (one file per figure). For guidance, download the 'Figure Guide PDF': (<https://www.embopress.org/page/journal/17574684/authorguide#figureformat>).
- 3) A .docx formatted letter INCLUDING the reviewers' reports and your detailed point-by-point responses to their comments. As part of the EMBO Press transparent editorial process, the point-by-point response is part of the Review Process File (RPF), which will be published alongside your paper.
- 4) A complete author checklist, which you can download from our author guidelines

(<https://www.embopress.org/page/journal/17574684/authorguide#submissionofrevisions>). Please insert information in the checklist that is also reflected in the manuscript. The completed author checklist will also be part of the RPF.

6) It is mandatory to include a 'Data Availability' section after the Materials and Methods. Before submitting your revision, primary datasets produced in this study need to be deposited in an appropriate public database, and the accession numbers and database listed under 'Data Availability'. Please remember to provide a reviewer password if the datasets are not yet public (see <https://www.embopress.org/page/journal/17574684/authorguide#dataavailability>).

8) We would also encourage you to include the source data for figure panels that show essential data. Numerical data should be provided as individual .xls or .csv files (including a tab describing the data). For blots or microscopy, uncropped images should be submitted (using a zip archive if multiple images need to be supplied for one panel). Additional information on source data and instruction on how to label the files are available at

9) Our journal encourages inclusion of *data citations in the reference list* to directly cite datasets that were re-used and obtained from public databases. Data citations in the article text are distinct from normal bibliographical citations and should directly link to the database records from which the data can be accessed. In the main text, data citations are formatted as follows: "Data ref: Smith et al, 2001" or "Data ref: NCBI Sequence Read Archive PRJNA342805, 2017". In the Reference list, data citations must be labeled with "[DATASET]". A data reference must provide the database name, accession number/identifiers and a resolvable link to the landing page from which the data can be accessed at the end of the reference. Further instructions are available at

13) Author contributions: the contribution of every author must be detailed in a separate section (before the acknowledgments).

14) A Conflict of Interest statement should be provided in the main text.

**** Reviewer's comments ****

Referee #1 (Comments on Novelty/Model System for Author):

N/A

Referee #1 (Remarks for Author):

In the research paper entitled "clinical grade ACE2 as a universal agent to block SARS-CoV-2 variants", Wirnsberger et al show in vitro efficacy data of recombinant human soluble ACE2 which protects host cells in two SARS-CoV-2 infection models. This is performed with several SARS-CoV-2 variants of concern (VOC).

In addition, affinity purified spike proteins of different variants are tested for binding affinity to clinical grade ACE2. Interestingly, affinity increases in S-proteins derived from VOCs with elevated infectivity.

This finding is of relevance since ACE2 can function as a SARS-CoV-2 neutralizing agent that is functional in many or all circulating VOCs. This may be an advantage over other therapeutic cell entry inhibitors such as neutralizing antibodies.

Nevertheless, I have some major and minor comments:

Major comment:

1. As stated above, the fact that ACE2 may function as a SARS-CoV-2 neutralizing agent independent from the VOC circulating in a specific region may be an interesting feature. However, in my view, this point is discussed and presented in a biased fashion especially when comparing ACE2 with neutralizing antibodies. The paper reads in large parts like a marketing leaflet promoting advances of drug A over drug B. This is of relevance since the authors hold patents on use of ACE2 for treatment of various diseases. However, EMBO MM is scientific journal in which pros and cons of different approaches should be discussed in an unbiased way. This should include mentioning of the facts that, in contrast to clinical grade ACE2, neutralizing SARS-CoV-2 antibodies have proven activity in several large clinical trials. Even some of the first generation ABs are also active against VOCs (e.g. casivirimab/imdevimab is active against the Delta variant and applied daily in COVID-19 patients). Novel 2. and 3. generation neutralizing antibodies with very broad activity against VOCs are in the pipeline or will be approved for clinical use. In parallel, therapeutic antibodies may have other, primarily pharmacokinetic advantages over ACE2: low IC50, very long half-life in humans. Half-life of ACE2 is relatively short compared to antibodies. Antibodies need to be applied as single shot, ACE2 most likely daily for several days. This should be discussed!

2. Last page of the discussion: here it is mentioned that use of monoclonal antibodies will drive viral evolution and generation of more VOCs. This is a very bold statement. I am not aware of any data supporting this statement. No citation for this provided either. Either provide data, citations or delete.

3. I don't see the point in a back-to-back publication of the two submitted papers. The in vitro study discussed here is relatively short and could easily be fused with the in vivo study Shoemaker et al.

4. Fig. 3A: why were different MOIs and infection times used for the different VOCs? Higher MOI and longer incubation times for VOCs compared to WT virus should have major impact on the outcome of infection experiments. In the methods section it reads: Suitable MOIs were optimized previously for each cell line and virus using serial dilutions and staining for SARS-CoV-2

nucleoprotein. What does this mean? Optimized in which sense?

5. The binding affinity studies were performed with the Delta variant which is circulating in large parts of the world causing 3rd and 4th COVID-19 waves in several countries. However, this variant was not tested (or date not presented) in the cell culture based assays (Fig. 3 and 4). Please comment.

6. Fig. 3: some of the curves may not be used for proper IC50 calculations. 3B Alpha variant has no bottom plateau. 3D Reference strain has no top plateau. Very large standard deviation in B1, B1-427, P2. Please comment.

Minor:

1. In Fig. 1c there seems to be a negative control or blank missing. Was there any background that was subtracted from the values?

Referee #2 (Remarks for Author):

Wirnsberger et al describe binding and neutralization capacity of clinical grade recombinant human soluble ACE2 (APN01) against SARS-CoV-2 VOCs. The authors show higher APN01 affinity and neutralization potency against several VOCs compared to the reference strain (2019-nCoV). Results were validated by two different laboratories. This study is supportive of the further clinical exploration of APN01 in COVID-19. However, this manuscript would require a substantial revision to address important points listed here below.

Major points:

- 1) In general, the higher affinity for VOC carrying N501Y was already reported by other studies. Affinity for kappa and delta was not shown to be improved in other studies, and in general the higher transmission rate of delta is attributed to a large extent to the P681R mutation in S2 and possibly other mutations in non-S viral genes. The fold change presented in Figure 1D (and also as manifest in Figure 1E for ref strain vs delta) shows only a 2-fold change for delta and kappa. Authors should tone down the Abstract in saying that the increased affinity of ACE2 is true for all VOCs. Of note, other mutations in S (even outside of RBD) may govern the dynamics of RBD opening enhancing the access to ACE2 in the context of the native trimer.
- 2) Another important topic that would merit a more insightful commenting is the paradox the protective role of therapeutic sACE2 versus the finding that high sACE2 is associated with more disease severity and fatality.
- 3) Paper from Zhang et al. Cell Discovery (2021)7:65 should be quoted. Authors should provide in Discussion a perspective on all ACE2 based therapeutics in development for COVID-19 and describe how this approach would differentiate from others.
- 4) Can the authors provide rationale for having used the RBD mutants reported in fig.1A-C? Apart from the aminoacidic substitutions highlighted in red in fig.1A that are present in several VOCs, we could not find the relevance of the others.
- 5) Fig. 1C: ELISA binding experiments were carried out using a single concentration of RBD variants and results reported as OD values compared to RBD-WT. While almost all RBDs tested showed higher binding (OD values >1), RBD-WT exhibited very low OD value (around 0.3) in these settings. Could the authors provide the background level of the ELISA assay and describe the method in more details?
Comparison between EC50 values extrapolated from a dose response binding curves would be a more accurate way to show these results.
- 6) Fig. 2B: please check the spike mutations listed for Beta and Gamma variants.
- 7) Delta variant was not tested in Figure 3. Considering this is currently the circulating virus Authors should provide neutralization data also against delta virus.
- 8) Fig. 3A: please clarify why the authors used different viral inputs and time of incubation prior to readout for the variants tested. Could these inconsistent settings explain the different neutralization potency shown for B.1.427 and B.1.429 variants despite having the same set of RBD mutations?
- 9) Fig. 3C: please explain why the IC50 values reported in the table not always match the neutralization curves shown in Fig.3B and 3D.
- 10) Fig 3: APN01 neutralization potency using a more physiological Calu-3 cell system seems to be lower compared to results obtained with Vero-E6 cells (e.g., Gamma, Alpha, Beta and P.2). Could the authors provide an explanation for these results?
- 11) Fig. 4C: could the authors clarify whether IC50 and IC90 values reported are calculated from 3 serial dilutions shown in panel A and B? If this is the case, a higher number of serial dilutions would be needed to extrapolate accurate values. Could the authors verify the statistical significance reported in panel A (level of inhibition of Alpha and Beta variants)?
- 12) In the text, the authors claim that "Combined with the increased binding affinity, changes in these kinetic parameters might contribute to the enhanced infectivity of VOC". Faster on-rate and lower off-rate both influence binding affinity and cannot be considered as independent parameters. If the authors refer to results in Fig. 1C, they should use the term "avidity" rather than "affinity".

Minor point:

Fig. 1A: Please match the aminoacidic substitution numbering with the order of display.

Referee #3 (Comments on Novelty/Model System for Author):

The virology is not appropriately described in this paper. Figure legends lack key details that are needed to interpret the data. As all antiviral studies appear to involve pretreatment of infectious virus with APN01, the effective multiplicity of infection (MOI) will vary by APN01 dose. Thus, the use of the term "MOI" is accurate when discussing these assays.

Referee #3 (Remarks for Author):

Clinical grade ACE2 as a universal agent to block SARS-CoV-2 variants by Wirnsberger et al. describes the antiviral activity of clinical grade recombinant human soluble ACE2 (APN01). It is written clearly. It is not appropriate to comment extensively on results from the "accompanying manuscript" as they are not presented in this manuscript, are not published, and thus cannot be evaluated appropriately by this reviewer.

1. The importance of virus dose or MOI in the assays described in this paper are not sufficiently described. If all studies described in this paper involve "pretreatment" of APN01 and virus for an hour prior to infection, MOI is not an appropriate descriptor. See below.
2. Figure 4. It is difficult to appreciate the antiviral activity of APN01 because the data is expressed as infection level percentage compared to mock. The actual titer data should be shown here to better understand the magnitude of the viral load drop and the potency of the virus titer reduction.
3. Figure 4. This figure describes pre-incubation of APN01 and SARS-CoV-2. Preincubation is not mentioned in the Figure Legend. This is really important information to interpret the data. The labeling of the figure with "MOI 0.01" and "MOI 1" is misleading as it implies to the virology community that the cells for all dilutions of APN01 were exposed to uniform amounts of infectious virus per MOI. This would not be an issue if cells were infected for an hour at a given virus dose and then exposed to dilutions of APN01. In the assay described in this paper, MOI, the ratio of infectious particles to the number of cells infected, varies with APN01 dose. Thus, rather than report MOI, it is more appropriate to report that X number of infectious particles were incubated with a dose response of APN01 for 1hr prior to infection.
4. Figure 4. This assay is really reporting the stoichiometry of APN01 binding to virus particles. At certain amounts of virus and APN01, the system is saturated where all particles are bound to APN01 prior to infection and this is titrated away as APN01 concentrations are decreased. What happens if you infect cells for an hour, wash away input virus not bound to cells and THEN add dilutions of APN01? If done in this manner infecting with a low MOI, you should be able to stop spread of infection and the antiviral effect would be magnified over time as the infection spreads in the untreated and conditions of APN01 that are insufficient to block spread. What if you infect cells at a higher MOI and then add APN01?
5. Given the reported IC50 and IC90 values in this paper, how do these data fit with what you know about dog/human PK data for the inhaled route?
6. The antiviral activity of aerosolized APN01 should be evaluated in human primary airway epithelial cells. This would provide important data on the ability of aerosolized APN01 to impact and ongoing SARS-CoV-2 infection in human cells.

Point by point reply to Referee #1

In the research paper entitled "clinical grade ACE2 as a universal agent to block SARS-CoV-2 variants", Wirnsberger et al show in vitro efficacy data of recombinant human soluble ACE2 which protects host cells in two SARS-CoV-2 infection models. This is performed with several SARS-CoV-2 variants of concern (VOC). In addition, affinity purified spike proteins of different variants are tested for binding affinity to clinical grade ACE2. Interestingly, affinity increases in S-proteins derived from VOCs with elevated infectivity. This finding is of relevance since ACE2 can function as a SARS-CoV-2 neutralizing agent that is functional in many or all circulating VOCs. This may be an advantage over other therapeutic cell entry inhibitors such as neutralizing antibodies. Nevertheless, I have some major and minor comments:

We thank the referee for the positive comments on the importance of our work. We have now added data on Omicron in the revised paper to expand the notion that ACE2 can indeed neutralize all relevant current VOCs. We think this is particularly important because Omicron appears to show an altered TMPRSS2 cleavage efficiency and altered tropism, resulting in changed rates of infection in different cell types, when compared to the other VOCs, making it paramount to show that ACE2 inhibition still works even in this scenario. We have also addressed all comments, please see detailed responses below, confirming and strengthening our conclusions.

Major comment:

1. As stated above, the fact that ACE2 may function as a SARS-CoV-2 neutralizing agent independent from the VOC circulating in a specific region may be an interesting feature. However, in my view, this point is discussed and presented in a biased fashion especially when comparing ACE2 with neutralizing antibodies. The paper reads in large parts like a marketing leaflet promoting advances of drug A over drug B. This is of relevance since the authors hold patents on use of ACE2 for treatment of various diseases. However, EMBO MM is scientific journal in which pros and cons of different approaches should be discussed in an unbiased way. This should include mentioning of the facts that, in contrast to clinical grade ACE2, neutralizing SARS-CoV-2 antibodies have proven activity in several large clinical trials. Even some of the first-generation ABs are also active against VOCs (e.g. casirivimab/imdevimab is active against the Delta variant and applied daily in COVID-19 patients). Novel 2. and 3. generation neutralizing antibodies with very broad activity against VOCs are in the pipeline or will be approved for clinical use. In parallel, therapeutic antibodies may have other, primarily pharmacokinetic advantages over ACE2: low IC50, very long half-life in humans. Half-life of ACE2 is relatively short compared to antibodies. Antibodies need to be applied as single shot, ACE2 most likely daily for several days. This should be discussed!

We apologize if the text was biased, that was certainly not our intention and, the referee is absolutely correct, the data need to be presented in a measured and critical way acknowledging the great work of others. As stated in our manuscript, the advantage of using recombinant versions of ACE2 is its inherent robustness towards novel variants of SARS-CoV-

2. As has unfortunately become obvious in the months since the initial submission, novel variants with the potential to escape neutralizing antibodies (both monoclonals, cocktails of monoclonals, and vaccine induced antibodies) are indeed a problem and require universally active therapeutic modalities. While our manuscript deals with one approach (the recombinant ecto-domain of ACE2) multiple other ACE2 based therapeutics are being developed and are of course referenced in our manuscript. This includes fusion proteins, as well as affinity optimized binders that address the pharmacodynamic and pharmacokinetic issues raised by the reviewer regarding a direct comparison with monoclonal antibodies. As has been observed before and is now especially prevalent for the Omicron variant, antibody-based therapeutics are vulnerable to mutational escape due to their smaller binding footprint in comparison to the Spike/APN01 interaction interface. Additionally, Omicron shows an altered dependency on the protease TMPRSS2, though a recent paper in Nature actually claims that inhibition of this protease is a pan-SARS-CoV-2 therapy. Considering these shifts and the potential of further alterations in protease usage and perhaps dependency we agree with the referee “*that one has to be careful what one claims*”. We clearly do not want to make a sales pitch or make wrong claims. However, as stated in the manuscript and in agreement with most researchers, SARS-CoV-2 cannot mutate out of ACE2 binding without affecting its cell entry.

2. Last page of the discussion: here it is mentioned that use of monoclonal antibodies will drive viral evolution and generation of more VOCs. This is a very bold statement. I am not aware of any data supporting this statement. No citation for this provided either. Either provide data, citations or delete.

We agree that this is a somewhat bold statement and have adapted the discussion accordingly. We would like to emphasize, however, that prolonged infection with SARS-CoV-2 has indeed been shown to favor the replication of SARS-CoV-2 variants with presumably immune-escape relevant mutations especially in the viral Spike protein. We believe that the extremely high number of mutations in the Omicron variant and its impact on viral neutralization in vaccinees and naturally infected subjects clearly indicates that the immune stimulatory properties and immunogenicity of the virus can change quite drastically, and that mass vaccination and therapeutics will affect the selection of novel viral variants, such as Omicron. Moreover, *in vitro* experiments have shown that incubation of SARS-CoV-2 with neutralizing antibodies results in escape mutants. We have however, as requested, changed that statement to not digress from our message “that ACE2 can block VOCs”.

3. I don't see the point in a back-to-back publication of the two submitted papers. The *in vitro* study discussed here is relatively short and could easily be fused with the *in vivo* study Shoemaker et al.

We have now added substantial new evidence to the *in vitro* study, including relevant novel variants of concern, as well as including data from vaccinee sera to test the effect of Omicron mutations on viral neutralization. Additionally, recombinant mouse ACE2 has been used for neutralization studies of Omicron to address the potential use of the mouse as model organism of SARS-CoV-2 infection studies, as well as highlighting the potential of novel animal reservoirs for SARS-CoV-2. Our data support and are in line with recent

evidence for SARS-CoV-2 variant spread into different animal species. Of note, based on the unfortunately rather negative review of the second paper, we have submitted that study to another Journal.

4. Fig. 3A: why were different MOIs and infection times used for the different VOCs? Higher MOI and longer incubation times for VOCs compared to WT virus should have major impact on the outcome of infection experiments. In the methods section it reads: Suitable MOIs were optimized previously for each cell line and virus using serial dilutions and staining for SARS-CoV-2 nucleoprotein. What does this mean? Optimized in which sense?

We thank the reviewer for this excellent comment. It is indeed true that the comparability of IC_{50} values is notoriously difficult, due to the experimental variation. The assays have been optimized to compare multiple therapeutics in their efficacy for one strain, not one therapeutic towards multiple strains. We have therefore repeated the experiments and have kept MOIs (or PFU, as stated in the table of Figure 3a) constant. Incubation times for the various VOCs are also comparable and have only been adapted to obtain ideal signals for the chosen read-out. We believe that this setup allows for a meaningful comparison of the various strains. Importantly, our new results obtained for this revised manuscript support the interpretation of the data presented in the original submission. We now clearly state in the revised material and methods section what “optimization” entails, also reflecting the necessity to obtain robust results for staining for the SARS-CoV-2 nucleoprotein. The kinetics of infection, production of viral proteins, and cell death need to be assessed to obtain the ideal point in time, which is indeed different for the various strains tested in this study.

5. The binding affinity studies were performed with the Delta variant which is circulating in large parts of the world causing 3rd and 4th COVID-19 waves in several countries. However, this variant was not tested (or date not presented) in the cell culture-based assays (Fig. 3 and 4). Please comment.

This has been corrected and Delta as well as Omicron have now been included in our biophysical, as well as cell-based assays. Figures 1, 2, 3, and 4 have been adapted accordingly in the revised paper.

6. Fig. 3: some of the curves may not be used for proper IC_{50} calculations. 3B Alpha variant has no bottom plateau. 3D Reference strain has no top plateau. Very large standard deviation in B1, B1-427, P2. Please comment.

The IC_{50} values were calculated from the depicted curves using GraphPad Prism software. As indicated, all strains were tested with a highest APN01 starting concentration of $10\mu\text{g/ml}$. Variances in the sensitivity to APN01 mediated neutralization between strains affected the slopes and amplitudes of the neutralization curves. While several of these curves appear truncated towards the upper or lower bound, all capture the point of 50% neutralization as well as 100% (or very close to 100%) neutralization. In figure 3D, unfortunately higher standard deviations have been an intrinsic feature of percent infections in lung Calu-3 cells,

something we always observe. Nevertheless, these curves still capture 0% to (near) 100% neutralization by APN01 in the tested tissue culture model. We hope that this is acceptable for the revised paper.

Minor:

1. In Fig. 1c there seems to be a negative control or blank missing. Was there any background that was subtracted from the values?

We apologize if that was unclear. Measurements without the addition of an RBD domain served as background and were subtracted, controlling for unspecific binding events observed in the absence of APN01/RBD interactions. This has now been added to the materials and methods section to clarify how the values were generated.

Point by point reply to Referee #2

Wirnsberger et al describe binding and neutralization capacity of clinical grade recombinant human soluble ACE2 (APN01) against SARS-CoV-2 VOCs. The authors show higher APN01 affinity and neutralization potency against several VOCs compared to the reference strain (2019-nCoV). Results were validated by two different laboratories. This study is supportive of the further clinical exploration of APN01 in COVID-19. However, this manuscript would require a substantial revision to address important points listed here below.

We thank the referee for the positive comments on the importance of our work. We have now added data on Omicron in the revised paper to expand the notion that ACE2 can indeed neutralize all current VOCs. We think this is particularly important because Omicron appears to show an altered TMPRSS2 cleavage efficiency and altered tropism, resulting in altered rates of replication in different cell types, when compared to the other VOCs, making it paramount to show that ACE2 inhibition still works even in this scenario. We have also addressed all comments, please see detailed responses below, confirming and strengthening our conclusions.

Major points:

1) In general, the higher affinity for VOC carrying N501Y was already reported by other studies. Affinity for kappa and delta was not shown to be improved in other studies, and in general the higher transmission rate of delta is attributed to a large extent to the P681R mutation in S2 and possibly other mutations in non-S viral genes. The fold change presented in Figure 1D (and also as manifest in Figure 1E for ref strain vs delta) shows only a 2-fold change for delta and kappa. Authors should tone down the Abstract in saying that the increased affinity of ACE2 is true for all VOCs. Of note, other mutations in S (even outside of RBD) may govern the dynamics of RBD opening enhancing the access to ACE2 in the context of the native trimer.

These are valuable points. Mutations outside the RBD domain have indeed been identified to be important for the infectivity and transmissibility of SARS-CoV-2 variants. To account for effects of non-RBD Spike mutations on the interaction with APN01, we also tested full-length Spike for its interaction with APN01. This does, however, not account for the full spectrum of observed effects, for example due to altered proteolytic processing, etc. We agree with the referee, that this needs to be pointed out explicitly. Regarding the observed affinities - we believe that a substantial body of literature has addressed these questions, the vast majority with comparable results that VOC mutations affect the affinity of Spike-ACE2 interactions. Our data support and extend previously published data and we stand by our observations and data, that VOC mutants affect and increase the observed affinities. It is of course critical to assess whether the observed effects (affinity increase) are biologically meaningful. This has been addressed in pseudotyping experiments in a study cited in our manuscript (Motozono et al. 2021), but other studies came to similar conclusions (Kim et al. 2021, BioRxiv): enhanced interaction with ACE2 in itself, as tested by introducing only RBD mutations or combinations of RBD mutations, is sufficient for increased infectivity mediated by the SARS-CoV-2 Spike protein. Although the main point of our manuscript is certainly the

potential to interfere with infection, not the increased infectivity provided by a higher affinity interaction with the cell entry receptor, we believe that the observed increased interaction is highly relevant for the proposed therapeutic approach because APN01 should and does indeed neutralize VOCs significantly better than the reference SARS-CoV-2 isolate.

2) Another important topic that would merit a more insightful commenting is the paradox the protective role of therapeutic sACE2 versus the finding that high sACE2 is associated with more disease severity and fatality.

ACE2 exists in two forms – a full-length form that is cell membrane bound and a shorter soluble form that is shed into body fluids and that normally circulates in the blood at very small amounts. There are reports that soluble ACE2 in plasma is moderately increased in patients with cardiovascular disease and also in patients with COVID-19. The significance of increased levels of soluble ACE2 in plasma in these conditions is not fully understood but likely reflects shedding of membrane bound ACE2 or cell-death mediated ACE2 release in these pathological conditions. The level of plasma soluble ACE2 in people with cardiovascular disease or patients with COVID-19 are at very low concentrations. Importantly, the observed increase in soluble ACE2 is minute, as compared to the amounts used for a treatment of COVID-19 infections, as highlighted by PK data obtained from previous clinical studies (Haschke et al. or Khan et al., referenced in our manuscript). This is especially relevant, as the primary mode of action for early disease intervention is largely based upon the neutralization of viral particles.

3) Paper from Zhang et al. Cell Discovery (2021)7:65 should be quoted. Authors should provide in Discussion a perspective on all ACE2 based therapeutics in development for COVID-19 and describe how this approach would differentiate from others.

We have now referenced 5 original articles describing alternative approaches highlighting ACE2 based therapeutics. Zhang et al. provides another excellent example that we have now added to the reference list. We thank the reviewer for pointing this out – in this crowded field we unfortunately simply overlooked this elegant manuscript.

4) Can the authors provide rational for having used the RBD mutants reported in fig.1A-C? Apart from the aminoacidic substitutions highlighted in red in fig.1A that are present in several VOCs, we could not find the relevance of the others.

During the early stages of the pandemic – before the most devastating VOCs were characterized (Delta and Omicron) – many variants with RBD mutations were described and deposited in databases. The initial thought behind the described mutant studies was therefore to assess the hypothesis whether these mutations also affect receptor interactions. As shown from the data, this indeed appears to be the case since the majority of the described and tested mutants affect this interaction. Given the strong impact of VOCs on the course of the pandemic, we put our focus on these variants in the revised manuscript and added these chronologically early experiments to Supplementary Figure 1. Since all the mutations are now highlighted in the RBD model (Figure 1) and trimeric Spike model (Figure 2) in the revised manuscript, we also removed the schematic (old Figure 1a), since it provides no added information. We hope that this is acceptable for the reviewer.

5) Fig. 1C: ELISA binding experiments were carried out using a single concentration of RBD variants and results reported as OD values compared to RBD-WT. While almost all RBDs tested showed higher binding (OD values >1), RBD-WT exhibited very low OD value (around 0.3) in these settings. Could the authors provide the background level of the ELISA assay and describe the method in more details? Comparison between EC50 values extrapolated from a dose response binding curves would be a more accurate way to show these results.

This is indeed a possibility, that we had considered. We however strongly felt that the biophysical characterization of this interaction using SPR would provide a deeper level of characterization and therefore did not perform further ELISA experiments. Background levels in the ELISA measurements were generally much lower than the reported OD value of 0.3. We are thus highly confident in these results. Of note, the background signals (i.e. the exact same experimental setup minus the addition of RBDs) were subtracted to derive reported OD values for the respective variants. This is now stated in the revised Materials and Methods section to clarify our experimental approach.

6) Fig. 2B: please check the spike mutations listed for Beta and Gamma variants.

We thank the referee for pointing this out. We have carefully checked the list of mutations and indeed corrected one mutation (P1). Beta and Gamma isolates from the NIAID did not contain the K417S/T mutation, however. Strains carrying the K417S/T mutations were tested at the Karolinska Institutet (Figure 4), to broadly cover circulating lineages and sub-lineages.

7) Delta variant was not tested in Figure 3. Considering this is currently the circulating virus Authors should provide neutralization data also against delta virus. This is indeed critical data at this stage of the COVID-19 pandemic. Both Delta as well as Omicron data have now been added to the manuscript in the revised Figures 3 and 4, as well as the relevant Supplementary Figures.

8) Fig. 3A: please clarify why the authors used different viral inputs and time of incubation prior to readout for the variants tested. Could these inconsistent settings explain the different neutralization potency shown for B.1.427 and B.1.429 variants despite having the same set of RBD mutations?

We agree with the reviewer that this is an important point. This has been experimentally corrected: equal viral PFUs have now been used for repeat experiments. Importantly, these new data confirm the conclusions derived from the original experiments. Incubation times were based upon results from pilot optimization studies and should not affect the conclusions. Importantly, incubation times for the VOCs are very similar. Moreover, RBD mutations are not the only relevant factors for the kinetics of the viral infections once the cell has been infected. We therefore believe that our findings for the VOCs are indeed relevant for the therapeutic assessment of APN01 and any ACE2 based therapeutic in COVID-19. In the limitations of our study section in the discussion we have now added a statement: *“Additionally, different incubation times with viral nucleoprotein staining as experimental readout might affect the accuracy of the obtained experimental results.”* We hope that the reviewer can accept this clarification.

9) Fig. 3C: please explain why the IC50 values reported in the table not always match the neutralization curves shown in Fig.3B and 3D.

Embarrassingly enough a copy-paste error caused this problem. We thank the reviewer for spotting this error which has been corrected in the revised manuscript.

10) Fig 3: APN01 neutralization potency using a more physiological Calu-3 cell system seems to be lower compared to results obtained with Vero-E6 cells (e.g., Gamma, Alpha, Beta and P.2). Could the authors provide an explanation for these results?

In pilot experiments, considerable differences regarding the infectivity of various strains and cell lines, including differences in VeroE6 and Calu-3 cells, were observed. This has also been observed by others (e.g. Chu et al. 2020, Lancet Microbe). It is therefore expected that these observed differences might also result in shifts when it comes to the potency of ACE2-based therapeutics, and, indeed, other therapeutic classes as well. We have not systematically addressed this, but it is conceivable, that the different susceptibility to SARS-CoV-2 infection of the used cell lines and/or the expression of additional molecules regulating the infection process (for example protease expression) might contribute to these differences. Importantly, these shifts in potency are very much compatible with the hypotheses and interpretations as raised and discussed in our manuscript.

11) Fig. 4C: could the authors clarify whether IC50 and IC90 values reported are calculated from 3 serial dilutions shown in panel A and B? If this is the case, a higher number of serial dilutions would be needed to extrapolate accurate values. Could the authors verify the statistical significance reported in panel A (level of inhibition of Alpha and Beta variants)?

We thank the reviewer for this insightful comment. IC₅₀ and IC₉₀ values were indeed calculated using 3 serial dilutions. We agree with the assessment that a higher number of dilutions would allow for a higher accuracy in the determination of these values. Since these experiments were used to confirm experimental results presented in Figure 3 and the readout chosen does not easily lend itself to automation, we reduced the number of serial dilutions to 3. Due to the experimental approach chosen in experiments shown in Figure 3, a higher number of serial dilutions could be used, hence the presumably more accurate determination of IC values presented in the respective Table in Figure 3. We hope that this is acceptable to the reviewer. Statistical analysis was performed using one-way ANOVA.

12) In the text, the authors claim that "Combined with the increased binding affinity, changes in these kinetic parameters might contribute to the enhanced infectivity of VOC". Faster on-rate and lower off-rate both influence binding affinity and cannot be considered as independent parameters. If the authors refer to results in Fig. 1C, they should use the term "avidity" rather than "affinity".

We are sorry not to have clearly stated and described the detailed setup for this experiment. To simplify the interpretation of the experiments, we immobilized APN01 as ligand onto the sensor chip. In this setup, the dimeric structure of APN01 is not relevant, since RBD

molecules are monomers. In this case RBD / APN01 interactions are characterized by 1:1 binding which was then analyzed for kinetic constant determination by mathematical sensorgram fitting applying a Langmuir 1:1 binding model. K_D values calculated from on- and off- rates represent affinities in this case. Since Spike (trimeric molecule) / APN01 measurements are characterized by the binding of Spike to APN01 containing two binding sites, this has been taken into account in the calculations presented in Figure 2.

Concerning the interdependence of on- and off- rates and binding affinities we believe that on and off-rates can be considered as independent parameters in this experimental setup. This is in fact an advantage of target binding affinity determination by kinetic analysis using Biacore or similar instruments. Two compounds interacting with a target molecule by the same affinity (= equilibrium dissociation constant) can achieve an identical affinity by two completely different interaction processes, e.g. a) fast association and fast dissociation or b) slow association and slow dissociation. Although both compounds have the same target affinity, they will behave very different in biological processes or when applied as pharmaceutical drug.

In case of SARS-CoV-2, "RBD wise" the first VOC Alpha achieved its selective advantage by N501Y resulting in much slower off-rate (compared to Wuhan) presumably allowing the virus to attach longer to ACE2 to achieve viral cell entry. In the Delta variant, N501 remains unchanged, but the mutation T478K appears to increase the on-rate, i.e. the virus binds to its docking site much faster than Wuhan reference strain virus again presumably resulting in a selective advantage. VOC Omicron combines both hot spot mutations thus retaining the fast (Delta) on-rate and presumably improving the off-rate over the Delta variant via the N501Y mutation.

Minor point:

Fig. 1A: Please match the aminoacidic substitution numbering with the order of display.

As discussed above, the schematic in Figure 1a has been removed, since the rendering in the new Figure 1a provides essentially the same information with the added benefit of mapping the mutations to the actual surface sites on the protein.

Point by point reply to Referee #3

(Comments on Novelty/Model System for Author): The virology is not appropriately described in this paper. Figure legends lack key details that are needed to interpret the data. As all antiviral studies appear to involve pretreatment of infectious virus with APN01, the effective multiplicity of infection (MOI) will vary by APN01 dose. Thus, the use of the term "MOI" is accurate when discussing these assays.

(Remarks for Author):

Clinical grade ACE2 as a universal agent to block SARS-CoV-2 variants by Wirnsberger et al. describes the antiviral activity of clinical grade recombinant human soluble ACE2 (APN01). It is written clearly. It is not appropriate to comment extensively on results from the "accompanying manuscript" as they are not presented in this manuscript, are not published, and thus cannot be evaluated appropriately by this reviewer.

We thank the referee for the thoughtful and supportive comments on our manuscript. We have now added data on Omicron in the revised paper to expand the notion that ACE2 can indeed neutralize multiple VOCs, providing additional evidence for further clinical exploration. We think this is particularly important because Omicron appears to show an altered TMPRSS2 cleavage efficiency and altered tropism, resulting in altered rates of replication in different cell types, when compared to the other VOCs, making it paramount to show that ACE2 inhibition still works even in this scenario. We have addressed all other issues (MOI, improved Figure legends, better methods description) and we have rewritten the manuscript according to all suggestions confirming and strengthening our conclusions. Please see detailed response below.

1. The importance of virus dose or MOI in the assays described in this paper are not sufficiently described. If all studies described in this paper involve "pretreatment" of APN01 and virus for an hour prior to infection, MOI is not an appropriate descriptor. See below.

We thank the reviewer for this insightful comment. As detailed in our manuscript, we observed (somewhat expectedly) a stronger inhibitory effect with a lower number of infectious particles/a lower MOI. Following this reviewer's suggestion, an alternative assessment concerning the virus dose has been added to the discussion section of the revised manuscript. We now compared the different strains using the same conditions in terms of the number of infectious particles/MOI and concentration of APN01 proteins. These new experiments again support our conclusions on the enhanced potency of APN01 to inhibit VOC strains as compared to the reference strain. Concerning the MOI, please see answer below to remark number 3.

2. Figure 4. It is difficult to appreciate the antiviral activity of APN01 because the data is expressed as infection level percentage compared to mock. The actual titer data should be shown here to better understand the magnitude of the viral load drop and the potency of the virus titer reduction.

The assessment of antiviral activity (expressed as infection level percentage as compared to mock infection) is based upon viral RNA quantification and we assumed that the

presentation of the data we chose was most suitable to depict the inhibitory effect. Alternatively, it would of course also be possible to show the data in a different format (see below for an exemplary depiction). If this data presentation is deemed more suitable for the depiction of our data, we can of course add these panels as a supplementary Figure or, alternatively, replace the panels in the respective Figures to increase the clarity and interpretability of the data for the readers.

3. Figure 4. This figure describes pre-incubation of APN01 and SARS-CoV-2. Preincubation is not mentioned in the Figure Legend. This is really important information to interpret the data. The labeling of the figure with "MOI 0.01" and "MOI 1" is misleading as it implies to the virology community that the cells for all dilutions of APN01 were exposed to uniform amounts of infectious virus per MOI. This would not be an issue if cells were infected for an hour at a given virus dose and then exposed to dilutions of APN01. In the assay described in this paper, MOI, the ratio of infectious particles to the number of cells infected, varies with APN01 dose. Thus, rather than report MOI, it is more appropriate to report that X number of infectious particles were incubated with a dose response of APN01 for 1hr prior to infection.

We thank the reviewer for raising this important point. We have rewritten the figure legends as requested to reflect this experimental set-up. This difference was indeed the point of our inhibitory approach: the reduction of infectious particles by the blockade of viral Spike with APN01 so viral particles can no longer bind to the cell expressed endogenous ACE2 receptor. Adding APN01 after infection would have addressed a different question: secondary spreading of the virus, which is of course a highly relevant point in itself but complicates the assessment of the anti-viral effect, especially considering the multitude of differences observed for viral infectivity of the different strains tested – not only regarding the initial binding of the virus to cell expressed ACE2. Experiments addressing viral spreading have indeed already been performed in a previous publication of our laboratory (Monteil et al,

2020, Cell) for the reference strain and have also been addressed in our *in vivo* experiments with mouse adapted SARS-CoV-2 (Gawish et al, 2022, Elife), further validating our approach. We hope that this is acceptable.

4. Figure 4. This assay is really reporting the stoichiometry of APN01 binding to virus particles. At certain amounts of virus and APN01, the system is saturated where all particles are bound to APN01 prior to infection and this is titrated away as APN01 concentrations are decreased. What happens if you infect cells for an hour, wash away input virus not bound to cells and THEN add dilutions of APN01? If done in this manner infecting with a low MOI, you should be able to stop spread of infection and the antiviral effect would be magnified over time as the infection spreads in the untreated and conditions of APN01 that are insufficient to block spread. What if you infect cells at a higher MOI and then add APN01?

This is indeed an important question. These experiments have been performed in our previous paper (Monteil et al. Cell 2020). Additionally, *in vivo* experiments in mice showed that ACE2 can inhibit infection and disease when given after the infection – the earlier we give ACE2 the better the protective *in vivo* effects (Gawish et al. Elife 2022). As outlined above we stayed away from this experiment in our current manuscript because of differences in viral replication once the various SARS-CoV-2 variants have entered the cells, making comparisons of ACE2 inhibition difficult.

5. Given the reported IC50 and IC90 values in this paper, how do these data fit with what you know about dog/human PK data for the inhaled route?

This question cannot be addressed directly by the obtained PK data since this would require the assessment of APN01 concentrations in the fluid along the lung lining or in the epithelia-lining fluids of the upper respiratory tract/oral/nasal cavity, the entry sites of SARS-CoV-2. These measurements are notoriously difficult to perform. Approximations on lung deposition of APN01 would lead us to hypothesize, that concentrations reached at the relevant sites should be at least 10 times higher than the IC₉₀ (assuming a fluid volume of 100ml per lung). That is, concentrations of APN01 in the lung lining fluids should be sufficient for the neutralization of SARS-CoV-2, especially for the VOCs, against which the potency of APN01 is significantly increased.

The assessment for systemic delivery is of course more straightforward: systemic levels of well above 1µg/ml are reached and are compatible with viral neutralization once the virus reaches the bloodstream. Please see Haschke et al. and Khan et al, both referenced in our manuscript, for relevant pharmacokinetic data.

6. The antiviral activity of aerosolized APN01 should be evaluated in human primary airway epithelial cells. This would provide important data on the ability of aerosolized APN01 to impact and ongoing SARS-CoV-2 infection in human cells.

While this is indeed an interesting experiment, we would like to point out, that we have little doubt on the validity of our approach based upon the following observations:

1. Many cell types and indeed even organoid systems have already been tested (published by us and others) using the original Wuhan virus as well as earlier virus variants with always

the same results: soluble ACE2 blocks the infection in a dose dependent manner, supporting our conclusions.

2. Using our mouse adapted SARS-CoV-2 virus that triggers lethal pneumonia and mirrors severe COVID-19 in humans, we have shown that respiratory application of soluble human (the adapted virus still binds to human ACE2, this is shown in the preclinical dog study of the originally back to back submitted paper) and soluble mouse ACE2 (Gawish, 2022, ELife) can effectively prevent SARS-CoV-2 infection, disease and death, arguing for both the important role of ACE2 in infections of primary lung epithelial cells as well as validating this general therapeutic approach.

In light of these observations, we hope that the reviewer will accept our model system as appropriate to draw the conclusion that the soluble ACE2-based therapeutic approach is feasible and - most importantly and the primary message of our present study - robust towards current and presumably future variants of SARS-CoV-2.

10th May 2022

Dear Prof. Penninger,

Thank you for sending us your revised manuscript. We have now heard back from the two referees who agreed to evaluate your study. As you will see, the referees are satisfied with the modifications made and think that the study is now suitable for publication.

Before we can formally accept your manuscript, we would ask you to address the following issues:

1. Please provide up to 5 keywords and incorporate them into the main text.

2. In Materials and Methods

- include a statement that informed consent was obtained from all subjects and that the experiments conformed to the principles set out in the WMA Declaration of Helsinki and the Department of Health and Human Services Belmont Report.

- Please add a sentence "Experiments dealing with SARS-CoV-2 were performed in a ... laboratory under the approval of...".

3. We updated our journal's competing interests policy in January 2022 and request authors to consider both actual and perceived competing interests. Please review the policy <https://www.embopress.org/competing-interests> and update your competing interests if necessary.

Please use the heading "Disclosure statement and competing interests".

Please add the following statement "Josef Penninger is an EMBO Member. This has no bearing on the editorial consideration of this article for publication."

4. Reference format: Please list 10 co-authors of a paper before adding et al. to the reference list.

5. Appendix:

- Please bundle Appendix figures and their legends together in a single pdf called "Appendix". Remove Appendix figure legends from the main manuscript file.

- Please update the nomenclature in the manuscript text and Appendix to "Appendix Figure S ##".

- The three tables in the Materials and Methods need to be moved to the Appendix file and labeled as "Appendix Table S ##".

- Provide a Table of Content on the 1st page in the Appendix.

6. CRediT has replaced the traditional author contributions section because it offers a systematic machine readable author contributions format that allows for more effective research assessment. You are encouraged to use the free text boxes beneath each contributing author's name to add specific details on the author's contribution. More information is available in our guide to authors. Please also remove the author contribution section from the manuscript file.

7. Please add a formal "Data Availability" section (placed after Materials & Method). Since this study does not generate large-scale datasets, please include the following sentence in this section- "This study includes no data deposited in external repositories".

8. Every published paper now includes a 'Synopsis' to further enhance discoverability. Synopses are displayed on the journal webpage and are freely accessible to all readers. They include a short stand first (maximum of 300 characters, including space) as well as 2-5 one-sentence bullet points that summarize the paper. Please write the bullet points to summarize the key NEW findings. They should be designed to be complementary to the abstract - i.e. not repeat the same text. We encourage inclusion of key acronyms and quantitative information (maximum of 30 words / bullet point). Please use the passive voice. Please attach these in a separate file or send them by email, we will incorporate them accordingly.

9. The Paper Explained: EMBO Molecular Medicine articles are accompanied by a summary of the articles to emphasize the major findings in the paper and their medical implications for the non-specialist reader. Please provide a draft summary of your article, highlighting

- the medical issue you are addressing,

- the results obtained and

- their clinical impact.

This may be edited to ensure that readers understand the significance and context of the research.

Please refer to any of our published articles for an example.

10. Our data editors have seen the manuscript, and they have made some comments and suggestions that need to be

addressed (see attached file). Please send back a revised version (in track change mode), as we will need to go through the changes.

11. As part of the EMBO Publications transparent editorial process initiative (see our Editorial at <http://embomolmed.embopress.org/content/2/9/329>), EMBO Molecular Medicine will publish online a Review Process File (RPF) to accompany accepted manuscripts.

In the event of acceptance, this file will be published in conjunction with your paper and will include the anonymous referee reports, your point-by-point response and all pertinent correspondence relating to the manuscript. Let us know whether you disagree with this and if you want to remove or keep any figures from it before publication.

Please submit your revised manuscript within two weeks. I look forward to seeing a revised form of your manuscript as soon as possible.

Kind regards,
Jingyi

Jingyi Hou
Editor
EMBO Molecular Medicine

*** Instructions to submit your revised manuscript ***

To submit your manuscript, please follow this link:

<https://embomolmed.msubmit.net/cgi-bin/main.plex>

- 1) a .docx formatted version of the manuscript text (including Figure legends and tables)
- 2) Separate figure files*
- 3) supplemental information as Expanded View and/or Appendix. Please carefully check the authors guidelines for formatting Expanded view and Appendix figures and tables at <https://www.embopress.org/page/journal/17574684/authorguide#expandedview>
- 4) a letter INCLUDING the reviewer's reports and your detailed responses to their comments (as Word file).
- 5) The paper explained: EMBO Molecular Medicine articles are accompanied by a summary of the articles to emphasize the major findings in the paper and their medical implications for the non-specialist reader. Please provide a draft summary of your article highlighting
 - the medical issue you are addressing,
 - the results obtained and
 - their clinical impact.This may be edited to ensure that readers understand the significance and context of the research. Please refer to any of our published articles for an example.

6) For more information: There is space at the end of each article to list relevant web links for further consultation by our readers. Could you identify some relevant ones and provide such information as well? Some examples are patient associations, relevant databases, OMIM/proteins/genes links, author's websites, etc...

7) Author contributions: the contribution of every author must be detailed in a separate section.

8) EMBO Molecular Medicine now requires a complete author checklist (<https://www.embopress.org/page/journal/17574684/authorguide>) to be submitted with all revised manuscripts. Please use the checklist as guideline for the sort of information we need WITHIN the manuscript. The checklist should only be filled with page numbers where the information can be found. This is particularly important for animal reporting, antibody dilutions (missing) and exact values and n that should be indicated instead of a range.

9) Every published paper now includes a 'Synopsis' to further enhance discoverability. Synopses are displayed on the journal webpage and are freely accessible to all readers. They include a short stand first (maximum of 300 characters, including space) as well as 2-5 one sentence bullet points that summarise the paper. Please write the bullet points to summarise the key NEW findings. They should be designed to be complementary to the abstract - i.e. not repeat the same text. We encourage inclusion of key acronyms and quantitative information (maximum of 30 words / bullet point). Please use the passive voice. Please attach these in a separate file or send them by email, we will incorporate them accordingly.

You are also welcome to suggest a striking image or visual abstract to illustrate your article. If you do please provide a jpeg file 550 px-wide x 400-px high.

10) A Conflict of Interest statement should be provided in the main text

11) Please note that we now mandate that all corresponding authors list an ORCID digital identifier. This takes <90 seconds to complete. We encourage all authors to supply an ORCID identifier, which will be linked to their name for unambiguous name identification.

Currently, our records indicate that the ORCID for your account is 0000-0002-8194-3777.

Link Not Available

12) The system will prompt you to fill in your funding and payment information. This will allow Wiley to send you a quote for the article processing charge (APC) in case of acceptance. This quote takes into account any reduction or fee waivers that you may be eligible for. Authors do not need to pay any fees before their manuscript is accepted and transferred to our publisher.

Photos 400-800 DPI

*Additional important information regarding figures and illustrations can be found at

<https://bit.ly/EMBOPressFigurePreparationGuideline>. See also figure legend preparation guidelines:

<https://www.embopress.org/page/journal/17574684/authorguide#figureformat>

The system will prompt you to fill in your funding and payment information. This will allow Wiley to send you a quote for the article processing charge (APC) in case of acceptance. This quote takes into account any reduction or fee waivers that you may be eligible for. Authors do not need to pay any fees before their manuscript is accepted and transferred to our publisher.

***** Reviewer's comments *****

Referee #1 (Comments on Novelty/Model System for Author):

-/-

Referee #1 (Remarks for Author):

The revised manuscript now meets the high standard of EMBO MM. No further comments.

Referee #3 (Comments on Novelty/Model System for Author):

This paper is technically proficient. The agent APN01 is not new and is being repurposed thus from the perspective of novelty, this paper is rated Medium. It's hard to gauge medical impact, but this kind of intervention would have to be given very soon after infection to have an impact on disease course. If combined with a therapy with a different modality, perhaps a cocktail approach will yield optimal results. Thus, impact is rated medium.

Referee #3 (Remarks for Author):

The revised manuscript titled "Clinical grade ACE2 as a universal agent to block SARS-CoV-2 variants" by Monteil et al. describes the antiviral activity of clinical grade recombinant human soluble ACE2 (APN01). The revision addresses all of my formed comments sufficiently nice job.

The authors have made all requested editorial changes.

3rd Jun 2022

Dear Josef,

We are pleased to inform you that your manuscript is accepted for publication and is now being sent to our publisher to be included in the next available issue of EMBO Molecular Medicine.

We would like to remind you that as part of the EMBO Publications transparent editorial process initiative, EMBO Molecular Medicine will publish a Review Process File online to accompany accepted manuscripts. If you do NOT want the file to be published or would like to exclude figures, please immediately inform the editorial office via e-mail.

Please read below for additional IMPORTANT information regarding your article, its publication and the production process.

Thank you for submitting this work to EMBO Molecular Medicine.

Kind regards,
Jingyi

Jingyi Hou
Editor
EMBO Molecular Medicine

Follow us on Twitter @EmboMolMed
Sign up for eTOCs at embopress.org/alertsfeeds

*** ** IMPORTANT INFORMATION ** **

SPEED OF PUBLICATION

The journal aims for rapid publication of papers, using the advance online publication "Early View" to expedite the process: A properly copy-edited and formatted version will be published as "Early View" after the proofs have been corrected. Please help the Editors and publisher avoid delays by providing e-mail address(es), telephone and fax numbers at which author(s) can be contacted.

Should you be planning a Press Release on your article, please get in contact with embomolmed@wiley.com as early as possible, in order to coordinate publication and release dates.

LICENSE AND PAYMENT:

All articles published in EMBO Molecular Medicine are fully open access: immediately and freely available to read, download and share.

EMBO Molecular Medicine charges an article processing charge (APC) to cover the publication costs. You, as the corresponding author for this manuscript, should have already received a quote with the article processing fee separately. Please let us know in case this quote has not been received.

Once your article is at Wiley for editorial production you will receive an email from Wiley's Author Services system, which will ask you to log in and will present you with the publication license form for completion. Within the same system the publication fee can be paid by credit card, an invoice, pro forma invoice or purchase order can be requested.

Payment of the publication charge and the signed Open Access Agreement form must be received before the article can be published online.

PROOFS

You will receive the proofs by e-mail approximately 2 weeks after all relevant files have been sent to our Production Office. Please return them within 48 hours and if there should be any problems, please contact the production office at embopressproduction@wiley.com.

Please inform us if there is likely to be any difficulty in reaching you at the above address at that time. Failure to meet our deadlines may result in a delay of publication.

All further communications concerning your paper proofs should quote reference number EMM-2021-15230-V3 and be directed to the production office at embopressproduction@wiley.com.

Thank you,

Jingyi Hou
Editor
EMBO Molecular Medicine